# Occupational injuries and associated factors among sanitary workers in public hospitals, eastern Ethiopia: A modified Poisson regression model analysis

Sina Temesgen Tolera[1]*, Tesfaye Gobena[1], Nega Assefa[1], Abraham Geremew[1], Elka Toseva[2]

1 Haramaya University College of Health and Medical Sciences, Harar City, Ethiopia, 2 Faculty of Public Health, Medical University of Plovdiv, Plovdiv City, Bulgaria

* sina.temesgen@haramaya.edu.et

## Abstract

### Background

Occupational or work-related injuries are mostly common among hospitals' sanitary workers (SWs) in developing countries like Ethiopia. This is due to improper practiced of devices, unhygienic workplace, neglected and undermined risk factors, as well as due to lack of policy initiatives; but not studied well.

### Objective

The aim of the study was to assess the occupational injuries and its associated factors among SWs in public hospitals, eastern Ethiopia: A Modified Poisson regression Model Analysis.

### Methods

An institution-based cross-sectional study was conducted in eight public hospitals in eastern Ethiopia from May 2023 to August 30th, 2023. Out of fourteen hospitals, eight of them were selected randomly. Eight data collectors and 4 supervisors were assigned. Face-to-face interview was conducted. Eight hundred hospital SWs were recruited for the study. Occupational injury was measured using Boolean logic questionnaire either YES [1] or NO [0] for the last 12 months and the 7 days. Descriptive statistical was used for means, medians, standard deviations, and frequencies, proportions, and percentages. Modified Poisson regression was used to explore the relationship of outcome and independent variables. Accordingly, bi-variable analysis was performed to estimate unadjusted prevalence ratio (UPR). While, multi-variable model was used adjusted PR(APR) for those variables have significant values of p ≤0.20 at bi-variate analysis with confidence interval of 95% (CI:95%).

**Data Availability Statement:** All relevant data are within the manuscript.

**Funding:** The author(s) received no specific funding for this work.

**Competing interests:** The authors have declared that no competing interests exist.

## Result

Out of eight hundred nine SWs, 729(90.1%) were participated on the study. Self-reported occupational injuries among SWs in the last 12 months were 44.0% (95% CI: 40.4, 47.7). Of these, 92.2% (95%CI: 88.7,94.90%) and 7.8% (95%CI: 5.1, 11.3%) occupational injuries was reported from the cleaners and waste collectors, respectively. The model found that SWs those acquired diseases after recruited in the hospitals (APR:1.3;95%CI:1.1,1.6), those had sleeping disorder (APR:1.2;95%CI:1.0,1.), those had workload (APR:1.3; 95% CI:1.0, 1.8), those exposed with occupational hazards (APR:1.4; 95%CI:1.3, 1.7) were at the risk of occupational injuries as compared to their counter parts. Meanwhile, SWs those didn't get supervision (APR: 1.0;95%CI: 1.0, 1.2) and those non-adherence to personal protective equipment (PPE) (APR:1.3;95%CI:1.0,1.5) were more likely to at the risk of occupational injuries.

## Conclusion

The current study concluded that there was a high prevalence of occupational injuries among SWs in the current selected public hospitals. The study also found that non-compliant with PPE, work load, sleeping disorders, attitude towards workplace safety and unsupervised activities and working in high-risk environment tends to increase the risk for occupational injuries. In addition to occupational injuries the study found that SWs those acquired occupational diseases such as asthma, respiratory tract problems, allergy, infections, kidney problems and dermatology problems after recruited in hospitals.

## Introduction

Health care settings like hospitals are the most workplace hazardous if they are not properly cleaning and sanitized in the daily working time. As the result, hospitals workers particularly SWs have a potential to risk [1]. Hospital cleaners are more susceptible to infection than other professions in these places [2]. Because they clean latrines, collect solid waste and manage liquid waste every day, making it inevitable that hospital-borne infections will occur go out [3–5]. Also, they are the most marginalized, with their dignity and human rights violated, socially stigmatized, discriminated against members of society [5–8]. They are living with poverty in terms of economy [9], and they have low wages for their works performed as compared to other occupations [10]. They are often cited as characteristics of the many "sweatshops" which operate in developing countries due to risky working environment; unhygienic workplace, physical abuse, long working hours [11]. Millions of sanitation workers across the world particularly in the developing world of health care facilities are forced to work in conditions that endanger their health and lives [5]; because of labor intensive and most of them have to perform under time constraints increasing the physical and mental stress [12].

Poor health and safety service have resulted in occupational injuries among SWs. They are not only suffering or acquiring occupational injuries, but also they face several occupational outcomes such as MSDs respiratory issues, dermatology, gastroenteritis and mental health [13]. Of these, occupational injury is considered. It defined as self-reported job-related physical harm to body tissues caused by accident or stressors exposure [14]. While, accident is an unexpected occurrence, happening by chance, implies a random and uncontrollable event.

Therefore, an injury refers to any physical harm that a person suffers as a result of an accident [15]. It can occur due to sharp material, slip, broken glasses, fall, manual handling and hit by falling objects, blood or body fluid exposure by splash, blood with infectious waste [16–18], splintering the waste into pieces [19]. The needle stick, splash and sharps were accounted for 90% of the puncture incidents due to slip and fall, and hit and caught; 43% of SWs had allergy and irritation; 31% of puncture and cut resulted from being caught or hit by equipment and garbage handling in health settings [14]. In this study it is said to high if higher than 10% and lower if less than less than 10%, adapted from [20].

The most common of occupational related injuries among SWs could be abrasions, fractures, trauma, dislocation, bruises, burns, cuts, amputations, and other job- related diseases [21–23]. Besides, they are exposed to hepatitis viral infections due to contaminated unsafe disposal of sharps and needles [24, 25], toxic residual chemicals, metal containers and heavy metals that increase cost of injuries [26, 27]. Therefore, poor health and safety are the worst consequences among employee particularly among SWs [21–23]. This affects not only workers, but also damaged products and reputation, create legal issues, increase the medical costs, increase turnover, decrease productivity [27], it can lead to absenteeism [28].

It is difficult to know the exact number of workers who die or are injured annually as a result of poor work conditions due to lack of reporting system and protocol [29]. The true economic burden of injuries is likely to be even greater because many of cases are not reported to the workers' benefit. The determinants or associated or risk factors for occupational injuries among workers in health care facilities are type of work shift, temporal employment, long working hours, extreme temperatures, lack of PPE and poor safety culture, which have been reported to be associated with occupational injurie [30, 31]. The other studies also found that gender difference [14, 32]; marital status [32, 33]; work experiences [14, 32–34]; lack of training and supervision [34–38], education [34, 39], environment, job dissatisfaction and job stress [34, 40], behavioral factors [29, 38], less attention of institutions [36]; weak infection prevention and control practice and work load [36–38, 41]; bad social recognition [42, 43] were attributed factors of the injuries. However, there was lack of study on occupational injuries and its determinants among sanitary worker in eastern Ethiopia's public hospital. Therefore, the current study was to assess occupational injuries and determinants among SWs in these regions' public hospitals.

## Materials and methods

### Study description and period

Hospital based cross section study was conducted in eight public hospitals in eastern Ethiopia (Fig 1). The study was period was from May 30th, 2023 to August 30th, 2023. At the beginning, eastern Ethiopia was purposely selected from the part of Ethiopia. At stage one, eastern Ethiopia was divided into four states: Oromia state, Harari Regional State, Somali Regional state, Dire Dawa city administrative. The number of public hospitals in the eastern Ethiopia are 14 general and referral hospitals. Of these, eight of them were selected randomly. An equal probability allocated to each region. Haramaya University Hiwot Fana comprehensive specialized hospital (HUFCSH), Jugola general hospital (JGH), Dilchora referral hospital (DRH), Sabian general hospital (SGH), Jigjiga University Sheik Hassan referral hospital (JUSHRH), Karamara general hospital (KGH), Bisidimo general hospital (BGH) and Chiro general hospital (CGH). The report found from the liaison shows that the mean and standard deviation (Mean ± SD) by bed occupancy in eight hospitals was 269.5 ± 132.6 per a day. Meanwhile, mean ± SD for outpatient flow and inpatients in these selected hospitals were 154.4 ± 67.4 and 233.9 ± 125.4, respectively. As the result, a mean ± SD total of 8 outpatient flow and inpatients are

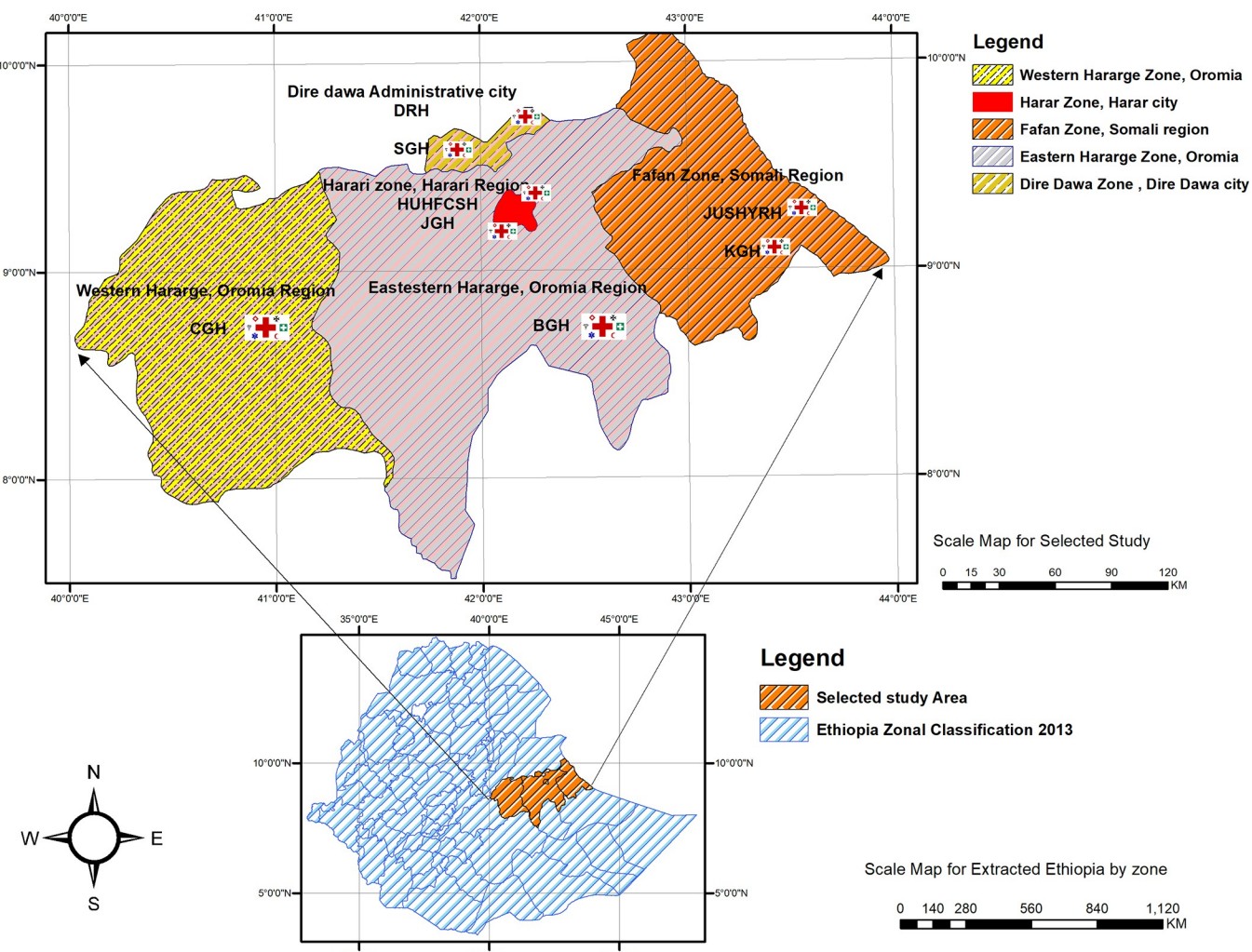

**Fig 1. Map of Ethiopia, selected eastern Ethiopia and selected public hospitals for the study created using ARCGIS from free access of Ethiopia GIS datasets [46].**

388.3 ± 190 [44]. As human power capital, about 5680 hospital staffs are working in the different department of these eight hospitals [45].

## Study population

All sanitary workers in all public hospitals in eastern Ethiopia were the source population. All cleaners and waste collectors working in the selected eight public hospitals were study population.

## Inclusion and exclusion criteria

The inclusion criteria were all SWs, namely cleaners and waste collectors including permanents, those on contracts and those who are outsourced, were recruited for study. All Infection Prevention and Control Focal person, or Environmental professional working in hospital [KI-1] and all sanitary workers' representatives or coordinators [KII-2] were included based relevance work experience as well closely working with SWs within their corresponding hospitals.

While, exclusion criteria were all hospital cleaners and waste collectors who were on annual, sick, and maternity leave during the study as well as those served less than one month were didn't include in the study.

## Sample determination

The sample size for burden of occupational injuries among SWs was estimated using single proportion formula, the reliability coefficient at 95% confidence interval (1.96), $p$ is the population proportion, $q$ is equal to $1-p$, and $d$ is the acceptable error (0.05). There is available on prevalence of occupational injuries (i.e. 52.9%) among hospital cleaners in Ethiopia [35]. The sample size (ni) become 383. Using design effect of 2.00, the sample size was 766. Adding with 5% contingency (38 individual), the final sample size was 804. This figure is approaching to all SWs in eight hospitals (N = 809). Thus, all units were recruited for the final study.

## Selection procedures

As illustrated above, eight public hospitals were selected at random from a total of fourteen public hospitals located throughout the regional states, with an equal probability allocated to each study location. Accordingly, a total of 234, 175, 82 and 318 SWs were eligible from HUHFCSH and JGJ (Harari regional state), BGH and CGH (Oromia regional state), DRF and SGH (Dire Dawa city administration) and from JUSHRH and KGH (Somali regional state), respectively (Fig 2).

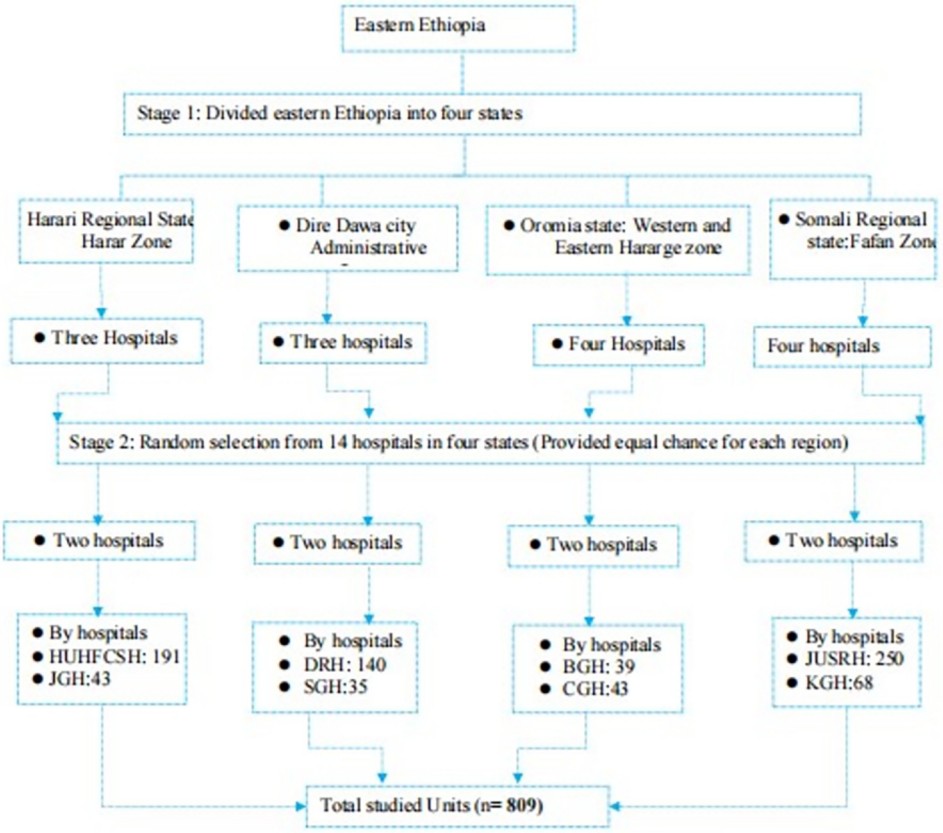

**Fig 2. Schematic representative of selection procedure of SWs, eastern Ethiopia, 2023.**

## Study variables

The dependent variable was occupation injuries, which could happen for the last 12-months different as a result of unsafe working condition, acts due to poor practice of occupational health and safety service in the public hospital settings, adapted from [32, 47]. While, the independent variables were socio-demographic variables: age, sex, education, experience, job categories, marital status, and monthly income. *Institutional variables*: Environmental and job satisfaction, workload, social recognition, and working hours. supply and availability of PPE, Occupational health and safety (OHS) training, IPC support. *Behavioral variables*: knowledge, attitude, PPE practice, alcohol consumption, cigarette smoking, chewing khat (the shrub Catha edulis (khat) leaves are chewed for their stimulant), job stress and sleeping disorder.

## Data collection methods

**Pattern of the study.** The pattern of the study was carried out into three types of shifts or job rotation for consecutive one month. Namely, first shift (Morning), second shift (Afternoon) and third shift (night). All hospital SWs with in the rotation of three shifts per week. The first shift starts at 7:00am and ends at 12:00am. The second shift starts at 1:00pm and ends at 5:00pm. The third shift starts at 12:00pm to 6:00am. This group works on schedule for two consecutive days and then rests for two days. By considering this, the questionnaires were administered (Morning: 9:00am-10:00am) for shift 1. For shift 2, the interview was done (Afternoon: 3:00pm-4:00pm]. The same procedure was done for shift 3 after 2 days.

**Data collectors.** Eight data collectors participated in data collection. All of them have Master degree in Occupational health and safety and environmental health and public health.

**Data collection tools.** Structured and standard closed questionnaires were used to collect quantitative data. Contents of the questionnaire include:

*Socio-demographic characteristics*: age, sex, educational status, work experience, job categories, marital status, and monthly income.

*Occupational injuries items*: *Structured* questions was developed for occupational injury adopted from relevant sources [32, 47] with required modification based on research objectives for last 12 months and in the last 7 days using Boolean Logic either YES [1] or NO[0]. Respondents were enforced to select cause and injured body parts afterward.

*Determinants /Associated Factors questions*: *E*ighteen standard questions were adapted for associated factors of occupational injuries [32, 47]. These includes: *Individual factors*: knowledge towards workplace risks dichotomous questions that measured by YES/NO, and changed into good (16–20 scores); 2: Fair (10–15 scores); and 3: Poor (< 10 scores). Also attitude towards workplace risks questions measured by 1–5 likert scale: Strongly agree to Strongly disagree. Then, using media, it was classified as favored if score is higher than 3.0, neutral if score is equals to 3.0 and unfavored if score is less than 3.0 *Behavioral Factors*: four questions using Boolean logic either YES [1] or NO [0] were prepared to assess the sleep disorders, heavy alcohol consumption, chewing Khat and smoke cigarette. *Institutional Factors*: four questions were prepared Boolean Logic either YES [1] or NO [0] to assess regular supervision of OHS and training, adequate PPE and work shift. Practice of personal protective equipment questions measured by Complaint and non-compliant). Infection prevention and control practice questions measured by Yes/No, and then classified into good (8–10 scores); 2: Fair (5–7 scores); and 3: Poor (< 5 scores). For conducting supervision that can be categorized as no supervision, sometimes and daily supervision and for the provision of post exposure prophylaxis, it has been classified as regular, some time and none of it. *Job satisfaction*: is a subjective response of study participants about their job whether it is pleasurable or not. *Work*

*environment satisfaction*: is mainly refers to the safety, comfort, and harmony of the objective hospital environment in which they work, excluding salary and promotion factors [48].

*Job stress symptoms*: A single item was used to measure stress symptoms at the SWs falls into a state of nervous tension, uneasiness, nervousness, anxiety, or sleeplessness at night because of certain psychological reasons [49]. Thus, it is a subjective response of respondents whether they feel stress due to the job or not. *Workload*, *work shift*: A single item for each factors were asked and then enforced to answer (YES [1]/NO[0]) adopted from [14]. Questions were prepared for the compliance with personal protective equipment, and then categorized as compliant and non-compliant using median. For conducting supervision that can be categorized as no supervision, sometimes and daily supervision, while for compliance with IPC practice, which was categorized as good practice, fair practice and poor practice. For the provision of post exposure prophylaxis, it has been classified as regular, some time and none of it.

## Key Informant Interview (KII)

*Key Informant Interview (KII)*: Two type of KII involved in this study from eight hospitals, namely: Eight (8) Infection prevention and control (IPC) focal persons (as KII-1) and 8 hospital sanitary workers' representatives (as KII-2). They selected purposely based on their expert and experience in the hospitals. All key informant interviewers had Bachelor Degrees and above. The feedback gathered from KII-1 and KII-2 in the means of qualitative data and quantitative data used to enrich the quantitative finding obtained from sanitary workers. Some key quantitative and quantitative information were provided for KIIs, separately as shows below.

**KII-1**: Fourteen (14) questions were prepared and all items were computed as *Severe* (S) X *Exposure* (E) X *Probability (P)* = (SEP) according to validated study [50]. Where, *Severity(S) is* the impact of the events, categorized as slight effect ranged from 1 to 5 (Score: 1), minor severe with absence illness (2), moderate injuries requiring hospital admission (3), major injuries and illness in permanent (4) and death (5). *Exposure(E) is the frequency* of events ranged from 1 to 5, categorized as very rarely, year (Score = 1) monthly (2), weekly (3) and daily (4), constantly, multiple times a day (5). *Probability(P) is the likelihood* of the events ranged from 1 to 5, categorized as chance to occurs is not expected through life of the work (Score = 1), it indicates chance to conceivable is between 10–30 of working years (2), chance between 1–10 working years (3), chance within a month (4), and chance to occurs daily to weekly (5). The output of SEP all items were rated as highly high (If 90–100% scores rated as 5), high (if 80–89% scores rated as 4), medium (if 60–79 rated as 3), low (if 50–59 score, rated as 2) and very low (If <50% score, it is rated as 1), adapted from [50].

**KII-2**: Sixteen (16) questions were prepared for SWs' representative (KII-2). The practice and applicability of the protocols' items were rated as used daily and available (score: 1), used weekly and not applied (0.75), used monthly and not sufficient (0.5) and used yearly and not sufficient (0.25), adapted from the validated study [51].

**Field Observation**: The field observation was conducted about the possible causes of occupational injuries among selected public hospitals. Observation checklist was used in order to assess utilization of gloves, mask, safety glasses/shield, apron/gown, and safety shoes by SWs [NSC, 52]. Observation for medical waste handling(MWH): The other observation check list was focused on MWH system in including safe sharp disposal system, adapted from [CDC, 53].

## Data quality

Standard and structure questionnaires was developed based on ideas from literature on compliance with the OHS conditions. To ensure quality of data, questionnaire was prepared in English (S1 File), then, translated into three local languages: Afan Oromo (S2 File), Amharic

(S3 File) and Somali (S4 File) to be used according to preferences of the speakers and regions. Moreover, for Key informant interviewers, English version was used (S5 File). Appropriate training was given for data collectors and supervisors. Reliability and validity data processing adapted from criteria [54, 55] used to be measured by Cronbach's alpha. In this study internal reliability coefficient was 0.93, which cutoff of 0.70 < "acceptable" [56]. The pretest was conducted on 5% of Haramaya general hospital's SW prior to the main study, which was not included in the main study.

## Data analysis

Coded quantitative data was double entered, validated, and cleaned using Epi-Data 3.1. The entered then exported to Stata 17MP version (StataCorp College Station, Texas, USA). The analysis followed three levels starting with descriptive exploration of the study variables, both the response variable and the predictors. Modified Poisson regression analysis was used for analytical statistics to explore outcome and predictors. The statistical models for binary outcomes suggested in literature include binary logistic regression model, modified Poisson regression model and log-binomial regression model [54–56]. The major weakness of logistic regression is that it tends to over-estimate the risk if the outcome of interest is not rare [57–59]. However, modified Poisson regression model is preferred in cross-sectional studies when the outcome of interest is not rare. Thus, the modified Poisson regression is approximate for the risk ratios or relative ratios better than binary logistic regression model [55]. Thus, bi-variate analysis model was used to perform Un-adjusted prevalence ratio (UPR). While, multi-variable model was used to analysis Adjusted PR (APR) for variables/predictors with significant values of $p < 0.20$ in the bivariate analysis. The results were deemed statistically significant at the two-tailed level with a 95% confidence interval and a p-value of 0.05. Moreover, the reliability of internal consistency of items was measured by Cronbach's alpha value. Accordingly, the value of was 0.93, which shows excellent internal consistency of items because it was higher than 0.70 [60]. Moreover, the multicollinearity of independence variables was measured by the variance inflation factor (VIF). Accordingly, VIF was 3.5, which was acceptable because it was less than cut point of 10 [61].

## Ethical statement

The study was approved by the Haramaya University College of Health and Medical Sciences, Institutional Health Research Ethics Review Committee (Ref ID: IHRERC/064/2023). Each Hospital were communicated about the study requesting their cooperation through a formal letter from the college. Each hospital has also given institutional consent for collecting health facility related data. All study participants were asked for their consent and sign on the consent form for their agreement to be part of the study (i.e. written type of consent). The confidentiality of the individuals was upheld in compliance with the Helsinki Protocol.

## Result

### Sociodemographic characteristics

Out of 809 SWs, 729 (90.1% were responded). Most of them were female (98.5%) and hospitals' cleaners (93.1%), permanent (97.3%) and worked shift one (49.4%). The mean ± SD for age, work experience, educational status, and monthly income were 34.4±7.6 years old, 6.7± 6.4 years, Grade 6.8±2.5, and 36.3±6.7 USD (United State dollar), respectively (Table 1).

**Table 1. Sociodemographic status of SWs in selected public hospitals in eastern Ethiopia, 2023.**

| Variables | Categories | Cleaners n (%) | Waste collectors n (%) | Total (N = 729) | | |
|---|---|---|---|---|---|---|
| | | | | Freq. (%) | | Mean ± SD |
| Gender | Female | 672(99.0) | 46(92.0) | 718 | (98.5) | |
| | Male | 7(1.0) | 4(8.0) | 11 | (1.5) | |
| Age | ≤ 24 years | 58(8.5) | 5(10.0) | 63 | (8.6) | |
| | 25–35 years | 326(48.0) | 24(48.0) | 350 | (48.0) | 34.4± 7.6 |
| | > 35 years | 295(43.5) | 21(42.0) | 316 | (43.4) | |
| Work experience | ≤ 2 years | 124(18.4) | 9(18.0) | 133 | (18.2) | |
| | 3–5 years | 271(39.9) | 17(34.0) | 288 | (39.5) | 6.7± 6.4 |
| | > 6years | 284(41.8) | 24(48.0) | 308 | (42.3) | |
| Education | ≤ Grade 4 | 148(21.9) | 12(24.0) | 160 | (22.1) | |
| | Grade 5–8 | 256(37.9) | 27(54.0) | 283 | (39.0) | 6.8±2.5 |
| | >Grade 8 | 271(40.2) | 11(22.0) | 282 | (38.9) | |
| Marital status | Single | 136(20.0) | 6(12.0) | 142 | (19.5) | |
| | Married | 467(68.8) | 39(78.0) | 506 | (69.4) | |
| | Others | 76(11.2) | 5(10.0) | 81 | (11.1) | |
| Monthly income salary (USD) | < 20.2* | 11(1.6) | 1(2.0) | 12 | (1.7) | |
| | 20.16–43.0** | 637(93.8) | 35(70.0) | 672 | (92.2) | 36.3±6.7 |
| | > 43.0 | 31(4.6) | 14(28.0) | 45 | (6.2) | |
| Employment type | Permanents | 660(97.2) | 49(98.0) | 709 | (97.3) | |
| | Contracts | 19(2.3) | 1(2.0) | 20 | (2.7) | |
| Type of shift | Shift 1 | 338(49.8) | 22(44.0) | 360 | (49.4) | |
| | Shift 2 | 242(35.6) | 20(40.0) | 262 | (35.9) | |
| | Shift 3 | 99(14.6) | 8(16.0) | 106 | (14.5) | |

Others: indicating separated and divorced; *(20.2USD = 1100ETB); **(2344ETB), salary classification based on national Job Evaluation and Grading [JEG, 2019] (Where 1Dollar ($) = 54.58 ETB, July 2023); *Shift 2 and 3* are working afternoon (workload somewhat rare)

## Occupational injuries

The overall self-reported occupational injuries among SWs in the last the 12 months was 44.0% (95%CI: 40.4, 47.7). Of these, 92.2% (95%CI: 88.7,94.9) and 7.8% (95%CI: 5.1, 11.3) were occupational injuries reported by cleaners and solid waste collectors, respectively. This does not imply that there were few solid waste collectors injured; rather, it was a result of the denominator from both of them being used extensively. However, based on their own denominator, it was 43.6% (95%CI: 39.9, 47.4) and 52.0% (95%CI: 35.5, 64.5) for cleaners (i.e. 296/679) and solid waste collectors (i.e 26/50), respectively. Likewise, the self-reported of occupational-related injuries among SWs for seven days was 2.6% (95CI:1.6, 4.0) (Table 2). Of this, contusion, cut wound, scratch, abrasion and laceration were accounted 71.3%, 54.8%, 53.6%%, 40.5% and 36.14%, respectively (Table 2).

## Bi-variable analysis model

Bi-variable Modified Poisson Regression model shows that being male SWs Un-adjusted Prevalence ratio(UPR: 1.9; 95%CI: 1.4,2.5), those acquired disease after recruited in the hospitals (UPR: 1.6;95%CI: 1.3,1.9], those developed MSD (UPR: 1.4;95%CI: 1.2,1.6), work load (UPR: 1.5;95%CI:1.3,1.8), working > 8hr/day (UPR: 1.3;95CI:1.1,1.6), those exposed with occupational hazards (UPR: 1.6;95%CI: 1.3,2.0) and had poor knowledge towards workplace risk

(UPR: 1.7,95%CI: 1.1,2.1) more likely to increase the risk of occupational injuries as compared to their counterparts (Table 3).

## Multi-variable analysis model

Multi-variable Modified Poisson Regression model shows that SWs those acquired diseases after recruited were adjusted Prevalence ratio (APR:1.3;95%CI:1.1,1.6), those had sleeping

**Table 2. Self-reported of injuries and causes of injuries among sanitary workers in public hospitals, eastern Ethiopia, 2023.**

| Items | Categories | Yes/No | Hospital cleaners Frequency (%) | Waste collectors, Frequency (%) | SWs: Total Frequency (%) |
|---|---|---|---|---|---|
| Had injured | Last 12 months | Yes | 296 (92.2) | 26(7.8) | 321(44.0) |
| | | No | 383(93.9) | 24(6.1) | 408(56.0) |
| | Last 7 days | Yes | 8(43.1) | 11(57.9) | 19(2.6) |
| | | No | 702(98.9) | 8(1.1) | 712(97.4) |
| Injury type* (n = 321) | Contusion | Yes | 212(72.1) | 17(68.0) | 229(71.3) |
| | | No | 82(27.1) | 10(32.0) | 92(28.7) |
| | Cut wound | Yes | 164(55.8) | 12(48.0) | 176(54.8) |
| | | No | 132(44.2) | 13(52.0) | 145(45.2) |
| | Scratch | Yes | 158(53.7) | 14(56.0) | 172(53.6) |
| | | No | 136(46.3) | 13(44.0) | 149(46.4) |
| | Abrasion | Yes | 117(39.8) | 13(54.2) | 130 (40.5) |
| | | No | 177(60.2) | 14(45.8) | 191(59.5) |
| | Laceration | Yes | 102(34.7) | 14(56.0) | 116(36.1) |
| | | No | 194(65.3) | 11(44.0) | 205(63.9) |
| | Dislocation | Yes | 86(29.3) | 13(54.2) | 99(30.8) |
| | | No | 211(70.8) | 11(45.8) | 222(69.2) |
| | Punctures | Yes | 103(35.0) | 13(52.0) | 116(36.1) |
| | | No | 193(65.0) | 12(48.0) | 205(63.9) |
| | Fracture | Yes | 101(34.4) | 13(54.2) | 114(35.5) |
| | | No | 195(65.7) | 11(45.8) | 207(64.5) |
| | Allergy | Yes | 98(33.3) | 12(48.0) | 110(34.3) |
| | | No | 198(66.7) | 13(52.0) | 211(65.7) |
| | Amputations | Yes | 80(27.2) | 11(47.8) | 91 (28.4) |
| | | No | 218(72.8) | 12(52.2) | 230(71.7) |
| Body part* injured (n = 321) | Finger | Yes | 157(53.4) | 17(68.0) | 174(54.2) |
| | | No | 139(46.6) | 8(32.0) | 147(45.8) |
| | Head | Yes | 164(55.8) | 16(66.7) | 180(56.1) |
| | | No | 133(44.2) | 8(33.3) | 141(43.9) |
| | Arms | Yes | 143(48.6) | 14(56.0) | 157(48.9) |
| | | No | 153(51.4) | 11(44.0) | 164(51.1) |
| | Teeth | Yes | 106(36.1) | 14(58.3) | 120(37.4) |
| | | No | 181(64.0) | 10(41.7) | 201(62.6) |
| | Legs | Yes | 94(32.0) | 15(60.0) | 109(34.0) |
| | | No | 202(68.0) | 10(40.0) | 212(66.0) |
| | Eyes | Yes | 77(26.2) | 12(52.2) | 89(27.7) |
| | | No | 221(73.8) | 11(47.8) | 232(72.3) |
| | Feet/toe | Yes | 84(28.7) | 12(50.0) | 96(29.9) |
| | | No | 213(71.3) | 12(50.0) | 225(70.1) |

(*Continued*)

**Table 2.** (Continued)

| Items | Categories | Yes/No | Hospital cleaners Frequency (%) | Waste collectors, Frequency (%) | SWs: Total Frequency (%) |
|---|---|---|---|---|---|
| Reasons for Injuries* (n = 321) | Sharp or needle Injuries | Yes | 198(67.8) | 18(72.0) | 216(67.3) |
| | | No | 94(32.2) | 7(28.0) | 105(32.7) |
| | Falls | Yes | 149(50.9) | 14(56.0) | 163(50.1) |
| | | No | 144(49.2) | 14(44.0) | 158(49.2) |
| | Hand tools | Yes | 157(53.8) | 12(50.0) | 169 (52.7) |
| | | No | 140(46.2) | 12(50.0) | 152(47.4) |
| | Slip | Yes | 100(34.3) | 10(40.0) | 110(34.3) |
| | | No | 196(65.8) | 15(60.0) | 211(65.7) |
| | Hit by falling objects | Yes | 115(39.4) | 8(33.3) | 123 (38.3) |
| | | No | 182(60.6) | 16(66.7) | 198(61.7) |
| | Splintering in waste | Yes | 78(26.7) | 7(28.0) | 85(26.5) |
| | | No | 218(73.3) | 18(72.0) | 236(73.5) |
| | Misuse of PPE | Yes | 92(28.0) | 8(33.3) | 100 (31.2) |
| | | No | 211(72.0) | 16(66.7) | 221(68.9) |
| | Fighting each other | Yes | 44(15.2) | 5(21.7) | 49 (15.3) |
| | | No | 246(84.8) | 18(78.3) | 272(84.7) |
| Work lost days(n = 321) | < 4days | | 153(96.2) | 6(3.8) | 159(49.5) |
| | > 4days | | 159(98.1) | 3(1.3) | 162(51.5) |
| AOD** | Yes | | 45(6.2) | 5(0.7) | 50(6.9) |
| | No | | 634(93.4) | 45(6.6) | 679(93.1) |

Asterisk * shows that there was more than one response (i.e. Multiple responses) from one individual; Asterisk ** shows that self-reported Acquired /developed occupational related diseases /AOD/ within public hospitals after recruited were MSDs accounted 50.7% and asthma and respiratory tract problems which was accounted 20.0%. In addition, they acquired allergy (8.0%), infections (12.0%), both bone fracture and dislocation (4.0%), kidney problems (34.0%) as well as dermatology problems (8.0%).

disorder (APR:1.2;95%CI:1.0,1.), those had workload (APR:1.3; 95%CI:1.0, 1.8), those exposed with occupational hazards (APR:1.4; 95%CI:1.3, 1.7) were at the risk of occupational injuries as compared to their counter parts. Meanwhile, SWs those didn't get supervision (APR: 1.0;95%CI: 1.0, 1.2) and those non-adherence with PPE practice (APR:1.3;95%CI:1.0,1.5) were more likely to at the risk of occupational injuries as compared to their counterparts. However, SWs those have neutral attitude towards workplace risk (APR: 0.5;95%CI: 0.3,1.0) more likely to reduce risk of occupational injuries by 50% as compared to unfavored attitude (Table 4).

## Goodness-of-fit of-model

The deviance of with the degree of freedom 716 on the other hand is 481.9. The dispersion parameter (value/DF), which indicates estimated deviance, is given in the Table 5. The value of the deviance good-of-fit divided by its degree of freedom gives 0.67, which is known as Dispersion parameter, which is approaching to 1. In addition, the value of Akaike's information criterion (AIC) and Bayesian information criterion (BIC) were 1153.9 and 1231.8, respectively (Table 5).

According to KII feedback, the possible occupational injuries among SWs could be 94.5% due to improper medical waste management at sources. In same manner, it was 91.6% due to needle stick, broken glass, plastic and sharp materials. In similar manner, 86.4%, 82.4% and 67.5% of occupational problems within the hospitals could be due to chemical detergents and

**Table 3. Bi-variable analysis model for predictors of occupational injuries among SWs in public hospitals in eastern Ethiopia, 2023.**

| Variables | Categories | Occupational Injuries (N = 729) | | UPR(CI:95%) | Sign. |
|---|---|---|---|---|---|
| | | Yes (n = 321) | No (n = 408) | | |
| Gender | Female | 312(43.5) | 406(56.5) | 1[Reference] | |
| | Male | 9(81.8) | 2(18.2) | 1.9 [1.4,2.5] | p = 0.001 |
| Work Experience | <3 Year | 52(39.1) | 81(60.9) | 1 | |
| | 3–6 Years | 123(42.7) | 165(57.3) | 1.1[0.9,1.4] | p = 0.01 |
| | >6 Years | 146(47.4) | 162(52.6) | 1.2[1.0,1.5] | p = 0.01 |
| Monthly Income salary | <20.2$ | 3(25.0) | 9(75.0) | 1 | |
| | 20.16–43.0 | 290(43.2) | 382(56.9) | 1.7[0.6,4.6] | p = 0.28 |
| | >43$ | 28(62.2) | 17(37.8) | 2.5[0.9,6.8] | p = 0.08 |
| Job rotation | Shift one | 176(48.9) | 184(51.11) | 1 | |
| | Shift two | 100(38.2) | 162(61.8) | 1.2[1.1,1.5] | p = 0.01 |
| | Shift three | 45(42.1) | 62(57.9) | 0.8[0.6,0.9] | p = 0.01 |
| Acquired Disease | No | 269(41.5) | 379(58.5) | 1 | |
| | Yes | 50(64.9) | 27(35.06) | 1.6[1.3,1.9] | p = 0.001 |
| MSD | No | 132(37.1) | 224(62.9) | 1 | |
| | Yes | 189(50.7) | 184(49.3) | 1.4[1.2,1.6] | p = 0.001 |
| Had sleep disorders | No | 245(46.1) | 287(54.0) | 1 | |
| | Yes | 76(38.6) | 121(61.4) | 1.2[1.0,1.5] | p = 0.08 |
| Work load | No | 219(39.0) | 343(61.0) | 1 | |
| | Yes | 100(59.9) | 67(40.12) | 1.5[1.3,1.8] | p = 0.001 |
| Working > 8hr/day | No | 221(40.3) | 328(59.7) | 1 | |
| | Yes | 98(54.4) | 82(45.6) | 1.3[1.1,1.6] | p = 0.001 |
| Smoking Cigarette | No | 286(43.1) | 377(56.9) | 1 | |
| | Yes | 35(53.0) | 31(47.0) | 1.2[1.0,16] | p = 0.096 |
| Had job stress | No | 202(39.0) | 316(61.4) | 1 | |
| | Yes | 119(56.4) | 92(43.6) | 1.4[1.2,1.7] | p = 0.000 |
| Bad social recognition | No | 215(51.9) | 193(61.3) | 1 | |
| | Yes | 199(48.1) | 122(38.7) | 1.2[1.0,1.5] | p = 0.013 |
| Occ. hazard exposures | No | 227(48.9) | 181(68.3) | 1 | |
| | Yes | 237(51.1)) | 84(31.7) | 1.6[1.3,2.0] | p = 0.001 |
| PPE compliance | Compliant | 143(41.9) | 198(58.1) | 1 | |
| | Non-compliant | 178(45.9) | 210(54.1) | 1.1[0.9,1.3] | p = 0.17 |
| Knowledge towards risk | Good | 227(42.0) | 314(58.1) | 1 | |
| | Fair | 66(44.3) | 83(55.7) | 1.1[0.9,1.3] | p = 0.61 |
| | Poor | 28(71.8) | 11(28.2) | 1.7[1,2.1] | p = 0.001 |
| Attitudinal towards risk | Favored | 141(45.3) | 170(54.7) | 1 | |
| | Neutral | 6(21.4) | 22(78.6) | 0.5[0.2,1.0] | p = 0.04 |
| | Unfavored | 174(44.6) | 216(55.4) | 1.1[0.9,1.2] | p = 0.54 |
| Conduct supervision | Daily supervision | 103(34.9) | 192(65.1) | 1 | |
| | Sometimes | 101(49.3) | 104(50.7) | 1.0[0.8,1.2] | p = 0.71 |
| | No supervision | 117(50.1) | 112(48.9) | 1.3[1.1,1.5] | p = 0.01 |
| IPC practice | Good practice | 93(35.2) | 171(64.8) | 1 | |
| | Fair practice | 136(50.8) | 132(49.3) | 0.9[0.8,1.1] | p = 0.39 |
| | Poor practice | 90(45.7) | 107(54.3) | 1.3[1.1,1.5] | p = 0.01 |
| PEP: Post Exposure Prophylaxis | Regular | 35(42.2) | 48(57.8) | 1 | |
| | Some time | 152(43.4) | 195(56.2) | 1.0[0.8, 1.2] | p = 0.71 |
| | Non-PEP | 134(44.8) | 165(55.2) | 1.3[1,1.5] | p = 0.07 |

**Table 4. Multi-variable modified Poisson regression analysis model for predictors of self-reported occupational injuries among SWs in public hospitals in eastern Ethiopia, 2023.**

| Variables | Categories | Occupational Injuries (N = 729) | | APR (CI:95%) | Sign. |
|---|---|---|---|---|---|
| | | Yes (n = 321) | No (n = 408) | | |
| Acquired Disease* | No | 269(41.5) | 379(58.5) | 1 | |
| | Yes | 50(64.9) | 27(35.06) | 1.3[1.1,1.6] | p = 0.01 |
| Sleep disorders | No | 245(46.1) | 287(54.0) | 1 | |
| | Yes | 76(38.6) | 121(61.4) | 1.2[1.0,1.5] | p = 0.04 |
| Work load | No | 219(39.0) | 343(61.0) | 1 | |
| | Yes | 100(59.9) | 67(40.12) | 1.3[1.0,1.8] | p = 0.02 |
| Occupational hazard exposures | No | 227(48.9) | 181(68.3) | 1 | |
| | Yes | 237(51.1)) | 84(31.7) | 1.4[1.3,1.7] | p = 0.01 |
| Conduct supervision | Daily supervision | 103(34.9) | 192(65.1) | 1 | |
| | Sometimes sup. | 101(49.3) | 104(50.7) | 1.0[0.7,1.5] | p = 0.92 |
| | No supervision | 117(50.1) | 112(48.9) | 1.1[1.0,1.2] | p = 0.01 |
| PPE compliance | Compliant | 143(41.9) | 198(58.1) | 1 | |
| | Non-compliant | 178(45.9) | 210(54.1) | 1.3[1.0,1.5] | p = 0.02 |
| Attitudinal towards risk | Favored attitude | 141(45.3) | 170(54.7) | 1 | |
| | Neutral attitude | 6(21.4) | 22(78.6) | 0.5[0.3,1.0] | p = 0.05 |
| | Unfavored attitude | 174(44.6) | 216(55.4) | 1.0[0.8,1.2] | p = 0.75 |

solvents and hazardous exposures, inappropriate utilization of PPE and insufficient PPE, respectively (Table 6).

## Assessment from sanitary representatives (KII-2)

The overall average of compliance with OHS and other welfare among selected public hospitals was 45.7%. The average of reporting system of injuries, initiation of post exposure prophylaxis

**Table 5. Goodness of fit test for modified Poisson regression model for the selected predictors of occupational injuries among SWs, 2023.**

| Criterion | Value | Df | Value/Df | Pr>Chi$^2$ |
|---|---|---|---|---|
| Model Test for only dependent variables | | | | |
| Deviance goodness-of-fit | 526.6 | 728 | 0.72 | p = 1.000 |
| Pearson goodness-of-fit | 408 | 728 | 0.56 | p = 1.000 |
| Dispersion parameter | | | 0.72* | |
| Model Test with independents variable | | | | |
| Wald squared | 78.71 | 12 | - | p<0.001 |
| Deviance goodness-of-fit** | 481.9 | 716 | 0.67 | p = 1.000 |
| Pearson goodness-of-fit | 404.1 | 716 | 0.56 | p = 1.000 |
| Dispersion parameter | | | 0.67** | |
| AIC | 1157.3 | 13 | - | - |
| BIC | 1217.0 | 13 | - | - |
| Log pseudo-likelihood | 565.7 | 12 | - | |
| Maximum Likelihood estimate | 0.77 | - | - | - |
| Pseudo R squared | 0.032 | 12 | - | - |

*Dispersion parameter at null = 0.72 and

**Dispersion parameter at multivariate = 0.67. Both values of dispersion parameter $\approx$1.00, indicating that the model shows equi-dipersion, which is acceptable Assessment from IPC as KII

**Table 6. Summary of IPC experts [KII-1] assessment about possible causes of injuries among SWs in public hospitals, eastern Ethiopia, 2023.**

| Type of possible causes of occupational injuries (Total Items score (N = 125) | N = 125 | % |
|---|---|---|
| Occupational injuries among SWs due to improper MW at sources | 118 | 94.4% |
| Occupational Injuries among SWs due to needle-stick and sharp materials | 115 | 91.6% |
| Occupational burns among SWs due to chemical detergents and solvents | 108 | 86.4% |
| Occupational problems among SWs due to inappropriate utilization of PPE | 103 | 82.4% |
| Occupational problems among sanitary workers due to insufficient PPE | 85 | 67.6% |
| Occupational injuries among sanitary workers due to lack of OHS training | 84 | 67.2% |
| OHS problems due to exposure to poor recognition for sanitary workers | 78 | 62.4% |
| Occupational injuries among SWs due to poor transportation of MWM | 69 | 55.2% |
| Develop occupational illness due to lack of post exposure prophylaxis | 68 | 54.4% |
| Exposure to biological hazards like HIV/AIDS, Hepatitis B and other pathogens | 65 | 52.0% |
| OHS problems due to poor practice of OHS and lack of its guideline | 65 | 51.6% |
| Occupational injuries due to poor infection prevention and control support | 67 | 53.2% |
| Occupational injuries due to poor passageway for transportation of medical waste | 46 | 36.4% |
| Occupational Injuries due to fall, slip, hit, caught equipment and materials | 27 | 21.2% |
| Average * | 78 | 62.6% |

*Indicating that there was non-compliance of occupational health and safety service

for the sanitary workers, hand hygiene after their work, medical waste management (MWM) practice in these hospitals were 25.0%, 25.0%, 43.8% and 40.6%, respectively. Also, according to KII-2, the government monitoring system on OHS service at their level in their corresponding hospitals was very quiet, almost none (Table 7).

## Field observation

Field observation was carried out during data collection period in both sanitary workers as well as within selected public hospitals. Accordingly, majority of SWs didn't use PPE properly following their responsibilities. The have the possibility of exposure with needle stick injuries as well as sharp materials as the result of improper waste segregation at source of each ward. The field observations (FO) revealed that six of them do not have a proper medical waste transportation flow, resulting in distributed wastes across the hospital owing to the antiquated design. Of these, code, 1, 3 and code 7 of selected hospital staffs including to SWs are the possibility of exposed with biological and chemical (FO, 2023).

## Discussion

The current study aimed to assess occupational injuries and its determinants among sanitary workers (SWs) in public hospitals, eastern Ethiopia. From 809 studied units, 729 of them answered the surveys. In case of non-response individuals, some of the people who chose not to participate in the study were told by the data collectors that: providing accurate information about ourselves would not guarantee any career progress or benefits from the study. Additionally, a few of them said that the hospital would suffer and that their pay would be affected if they revealed their embarrassing work injuries within the hospitals. Therefore, this imply as it is important to take serious consideration of occupational health and safety regulations, policies, and recommendations at public hospital about SWs workplace rights and obligations about workplace information.

**Table 7. KII-2 Feedback on existence OHS safety service including reporting system of occupational injuries and other welfare practice in public hospitals, eastern Ethiopia, 2023.**

| Items Type | Code of KII with the corresponding their hospitals (Code:01–08) | | | | | | | | Average (%) |
|---|---|---|---|---|---|---|---|---|---|
| | 01 | 02 | 03 | 04 | 05 | 06 | 07 | 08 | |
| Develop OHS Guidelines | | | | | | | | | 50.0 |
| Reporting system of injuries | | | | | | | | | 25.0 |
| Training for cleaners | | | | | | | | | 53.1 |
| Adequate supervision | | | | | | | | | 62.5 |
| Job rotation for SWs | | | | | | | | | 62.5 |
| Availability of PPE | | | | | | | | | 50.0 |
| Comfort-ability of PPE | | | | | | | | | 50.0 |
| PPE Utilization | | | | | | | | | 56.3 |
| Initiation of PEP | | | | | | | | | 25.0 |
| Confined room for workers | | | | | | | | | 50.0 |
| Confined Space for PPE | | | | | | | | | 53.1 |
| Hand hygiene after work | | | | | | | | | 40.6 |
| Well design path for MWM | | | | | | | | | 53.1 |
| OHS monitoring by gov't | | | | | | | | | 0.0 |
| Medical waste management | | | | | | | | | 43.8 |
| General OHS Initiative | | | | | | | | | 56.3 |
| | Average | | | | | | | | 45.7 |

| | | | | | | |
|---|---|---|---|---|---|---|
| • Well, supplied/used<br>• Daily used/applied<br>• Given Value = 1.00 | • Partially supplied<br>• weekly applied<br>• Value = 0.75 | • Not enough<br>• Monthly used<br>• Value = 0.50 | • Not enough<br>• Once/year<br>• Value = 0.25 | • Not designed<br>• Not applied<br>• Value = 0.00 | • Unaware<br>• Unknown<br>• Value = 0.00 | • Average<br>• 100% |

Regarding the outcome assessment, the high percentage of occupational injuries was reported (44%) among hospital SWs. As contrast, it is slightly higher than 35.9%, the finding obtained from Texas's public hospitals, USA [23]. The discrepancy may be explained by a lack of knowledge regarding the perception of workplace risk, which the current study found to be highly correlated with the frequency of occupational injuries among sanitary workers in hospitals. The difference may also be attributable to the educational attainment of sanitation workers, as Texas is a higher-income state than the low-income nations where the current study was carried out. The current study also slightly lower than 47.0% from another part of Ethiopia [62]. The disparity may result from the participant mix and data gathering procedures utilized in the prior study, which compared low-educated sanitary workers with highly educated healthcare workers, potentially leading to sample bias in that study as compared to this study. Moreover, the current study was lower than 56.0% that obtained from the Palestine [63] and also lower than 62.0% found Canadian province of British Columbia [14], 52.0% from Jordan [64], 55.0% obtained from Taiwan [65]. It does not imply that the sanitary workers working in these research areas had enough protection, enough training on health and safety, or understanding of workplace health and safety issues. However, the gap may be explained by sanitary staff' likely belief that it would be embarrassing to disclose occupational injuries because doing so would damage the hospital's reputation, even if they reported the problems honestly.

However, according to the feedback obtained from KII-1, 54.3% of hospitals' sanitary workers did not adhere to occupational health and safety regulations. This indicates that a certain proportion of occupational issues were brought on by unsatisfactory OHS care at the hospital. This assumption is also supported by finding obtained from KII-2 of the code 1, 2, 3 stated that,

"I do not know about occupational health and safety issue first because we did not have any awareness about it, as a result I couldn't say something about it." (Code 1, 2, 3 of SW-Rep,2023).

This report similar to the previous study [30], the key obstacles faced by SW in adhering to OHS principles were determined to be poor compensation, insufficient tools and equipment, poor supervision, harsh weather conditions, and abuse as well as assault [66]. According to report obtained from KII-1, the percentage of OHS training practice among selected public hospitals was 66.9% (Table 4). In addition to this, the feedback obtained from KII-2 shows that the mean percentage of health and safety training practice at selected public hospitals was 53.1% (Table 5). They reported to data collector as "We asked about the training to our respective team leaders (IPC experts), they responded us we haven't budget to do it" (SW-Rep, 2023). Of these, code 3 and code 5 of SW-Rep. reported that,

"Almost all of our staffs didn't get OHS training regarding to work character, on our right in workplace and utilization of PPE as well as how we are trying to prevent and control stressors like biological and chemical hazards that we exposing in our daily work".

According to KIIs' inputs, the mean percentage of compliance with OHS among the hospitals was 44.7% (Table 5). In addition, SW-Rep, code 1 report: "I had injuries on my right leg in the months of February, 2023, but I didn't get any compensation and treatment from the hospital because there are no occupational related injuries reporting system and management within the hospital,". As a result, I was out of my work for more than a month." Inline this issue, IPC experts (KII-1) and the SW-Rep (KII-2) from the corresponding hospitals confirmed to the fact that public hospitals paid little attention to or awareness of OHS services for their sanitary staffs.

The current study also identified the most reported body areas affected from injuries among sanitary workers namely cleaners and waste collectors in public hospitals. The descriptive result shows that the prevalence of head injuries among these group was 55.8%. This is the most common injury kind among others. It happened when they were cleaning and carrying something that extended beyond their head and fell on their head. This result was somewhat consistent with those of the public hospitals in Texas, USA [23].

The prevalence of sharp and needle stick injuries for occurrence of occupational injuries among sanitary workers in this study was 67.3%, higher that the study conducted in Texas, USA where needle trash cleaning and sharp object were accounted for 17% of the injuries [23] and higher that 52.9%, where more than half of the of cleaners in Tikur Anbassa specialized referral hospital had experienced injuries due to sharp and needle stick [49]. According to the KII-2 survey, 91.6% of sanitary workers could sustain or probabilities to have an occupational injury from a needle stick, shattered glass, plastic, or sharp object (Table 4). Inappropriate hand tools were potential causes of occupational injuries among sanitary workers shared about 50.1% of the cases. Also, falls also the other potential causes of occupational injuries among sanitary workers at public hospitals, was accounted 65%.

Medical waste handling and splitting up waste into pieces was shown to be the other cause of occupational injuries among these categories, accounting for 26.5% of cases. To confirm this evidence, the results of the KII-2 feedback showed that 94.5% of occupational injuries among SWs were thought to be the result of incorrect medical waste treatment at the source. Despite of fact that the above finding was less than 31% of SWs working in Columbia's public hospitals, where medical waste handling problems among these groups such as cleaners [14]. It does not imply that PPE was used appropriately by SWs in the research regions. However,

the majority of the study locations' SWs may not have reported PPE usage because they believed it would be embarrassing to do so. This might account for the disparity.

In this study, the typical type of occupational injuries among SWs were contusion (31.41%), cut wound (24.1%) and scratch (23.6%). The study found that the prevalence of percutaneous injury among cleaners was 46.0% [63]. The chance of accidents increasing among SWs due to low knowledge towards risk perception. As previously said, they are aware of occupational risks that may arise at the hospital, but they may not realize the seriousness of the risk as their counterparts do with poor knowledge [28].

A high frequency of absences due to illness for a short duration (less than 4 days) is one of the indicators of job dissatisfaction. In our study, this was observed in the group of hospital waste collectors. Some authors find a relationship between the frequency of absences due to illness and the level of monthly income, respect of experience at the workplace etc [28]. There is no uniform methodology worldwide regarding the definition of sickness absences as short or long according to the duration of time. Probably, this is also due to differences in the legislation of the different countries. The study found that short incantations of sickness absence these less than 7 days and as a long periods these above 7 days [67]. In most of cases long-term sickness absence usually defined as more than 4–6 weeks (or more than 28 days) of sickness absence [67]. Absences from work, including reasons of occupational injuries, can be an indicator of gaps in the organization of work. They enable us to evaluate the psycho-social and physical health of hospital SWs, and provide an opportunity to evaluate the policy of public hospitals towards these group of workers [67].

Modified Poisson regression of the final model shows that SWs those acquired occupational diseases after recruited in the hospitals were increased risk of occupational injuries by 1.3 times as compared to those didn't acquire any type of occupational disease (Table 4). According to current finding, the most self-reported occupational diseases among these groups was asthma and respiratory tract problems was accounted. In addition, they are also acquired allergy, infections, bone fracture and dislocation, kidney problems as well as dermatology problems.

The study found that the prevalence ratio of individual those had more likely to increase the risk of occupational injuries as compared to those haven't [68]. According to the model, there was a 1.2-fold increase in the probability of occupational injuries for SWs with sleeping disorders compared to those without (Table 4). This is because poor sleepers have reduced reaction times and trouble concentrating, with an increased likelihood of accidents or making costly mistakes, according to Nuffield Health [69].

The final model also demonstrates that SWs with hospital workloads had a 1.3-fold higher risk of occupational injuries compared to those without it (Table 4). This is because a spike in workload causes an increase in neuromuscular fatigue, and an increase in neuromuscular fatigue leads to an injury according to by Johann Windt report [70]. Moreover, SWs those exposed with occupational hazards were increased risk of occupational injuries by 1.4 times as compared to their counter parts (Table 4). The study found that predominant source that contributed for the occupational injury workplace hazards, namely physical, psycho-social, and work organization [71].

According to Modified Poisson regression shows that unsupervised SWs were more likely to increase the risk of injuries as compared to those supervised during their works. This is because workplace training is essential to safety since it reinforces appropriate processes and work practices that address systemic faults. Therefore, it is possible to view a lack of OHS, especially OHS monitoring, as a factor in an inadequate safety culture that causes workers to suffer from consequences related to their jobs [72]. The model also found that SWs those non-adhered to personal protective equipment practice were increased risk of occupational injuries

by 1.3 times as compared to those adhered to personal protective equipment (PPE) practice during their work (Table 4). The study other confirmed that the absence of and inappropriate PPE, hereinafter referred to as PPE non-compliance, are major causes of fatal and nonfatal occupational impairment at workplaces [73]. However, compared to those with an unfavorable attitude, SWs with a neutral attitude toward workplace risk are more likely to lower the likelihood of occupational related injuries by 50% times (Table 4). The study confirmed that low levels of perceived risk and estimation of ability were associated with a significant increase in risk of occupational related injuries [74]. Despite of fact that in this study, it is not seen to be statistically significant that positive attitudes of SWs regarding workplace risk are either more likely to lower the probability of injuries. Thus, there is no enough evidence to reject the null hypothesis about it.

Moreover, according to field observations majority of SWs neglected to wash their hands after their shifts ended in these public hospitals due to lack of hand-washing facilities. As the result they could be exposed to chemical or biological risks using this kind of approach, which is common in healthcare settings. The findings of the field observation also demonstrate that most of them failed to wear PPE appropriately in order to carry out their duties at work. In terms of medical waste management, field observations in eight selected public hospitals revealed that nearly seven of them did not appropriately separate and segregate medical wastes based on their characteristics and kind that leads to occupational impairments. Regarding to hospital environment, field observations revealed that six of them do not have a proper medical waste transportation flow, resulting in distributed wastes across the hospital owing to the antiquated design. Moreover, the possibility of occupational bio-hazards occurrence highly seen around floor of surgical and medical wards of code 1, code, 2, code 4 and code 6 of selected hospitals (Field Observation, 2023).

Regarding model of good-of-fit Modified Poisson regression discussion, the test was demonstrated before (to check model fit) and after (for further report of model). Accordingly, the criteria of goodness-of-fit for Modified Poisson regression are shown in Table 6. This Table shows that the values of the Pearson Chi-square (404.1) indicated that the standardized distances between the observed and expected responses. While, the value of deviance goodness-of-fit was 481.9, that is an indication individual contribution to the model deviance between the log-likelihoods of the saturated and fitted models is 481.9. Both values are not equal because modified Poisson is a counted regression models where their distributions are often skewed and non-normally distributed for counts regression models. Despite it is argued that the deviance goodness -of-fit typically follow more closely a normal distribution than the Pearson goodness -of-fit according to [75]. The deviance and Pearson Chi-squares value are lower than the degrees of freedom; the P-values for the deviance and Pearson Chi-squares are all larger than 0.05 (1.00, 1.0). The value of dispersion parameter was 0.67 for dependent variable and for those significant of independent variables, which is approaching to 1.00. This is indication or best fulfilled major assumption of Modified Poisson regression where equidispersion equals to 1.00 or dispersion parameter is 1. Therefore, overall conclusion, this result suggested that Modified Poisson Logistic regression was appropriate model for the current analysis.

## Strength and limitation

### Strength of the study

There are several strengths to this study. The current study produced good information along with experts about occupational injuries among sanitation workers in public hospitals, specifically in eastern Ethiopia that may serve as a future point of reference for similar research. All

populations were assessed with simple sampling procedures that reduce the possibility of sampling bias resulting from the selection of a sample of respondents from a diverse. Additionally, the study was able to estimate the components that are currently being studied as well as the prevalence of occupational injuries. Moreover, the research offers compelling evidence supporting the development of research hypotheses on occupational injuries among SWs.

### Limitation of the study

Despite this advantage, the cross-sectional study design used in this study makes it unable to look at changes over time or pinpoint the reason for the outcome. The relationships that have been established may be problematic to identify as the causes of occupational injuries because the prevalence of this study has not been reported in the researched locations. The study was aimed assess occupational injuries among sanitary workers for the last years, which can induce or vulnerable to recall bias. Another disadvantage is that the data might not match the information given because it was collected directly from sanitary staff members throughout the course of the prior year.

## Conclusion

The study concluded that a burden of occupational injuries among sanitary workers observed among selected public hospitals. This problem was as the result of poor knowledge and unfavorable towards workplace risk, lack of regular supervision, workload and sleep disorders among SWs. It is more prevalence as result of presence of additional musculoskeletal disorders and the presence of occupational hazards were significantly associated with the occurrence of occupational injuries. Therefore, the study advised that it is important to take serious consideration of occupational health and safety regulations, policies, and recommendations for the Federal ministry of Health and labor affairs. Also, hospital managers should take note of the occupational risks among SWs those are facing in their day-to-day activities and devise a mechanism to avert them with available resources.

## Supporting information

**S1 File. English version questionnaires.**
(PDF)

**S2 File. Afan Oromo version questionnaires.**
(PDF)

**S3 File. Ahmaric version questionnaires.**
(PDF)

**S4 File. Somali version questionnaires.**
(PDF)

**S5 File. Key informant interview English version questionnaires.**
(PDF)

## Acknowledgments

We would like to thank: the leadership of the Haramaya University College of Health and Medical Sciences, the administration of the eight public hospitals in Eastern Ethiopia: (HUHFCSH, JGH, DRH, SGH, JUSHRH, KGH, BGH and CGH). We would like to also acknowledged the data collectors as well as sanitary workers, who took part in the study.

## Author Contributions

**Conceptualization:** Sina Temesgen Tolera, Tesfaye Gobena, Nega Assefa, Abraham Geremew, Elka Toseva.

**Data curation:** Sina Temesgen Tolera, Tesfaye Gobena, Nega Assefa, Abraham Geremew.

**Formal analysis:** Sina Temesgeh Tolera, Tesfaye Gobena, Abraham Geremew.

**Funding acquisition:** Sina Temesgen Tolera.

**Investigation:** Sina Temesgen Tolera, Tesfaye Gobena.

**Methodology:** Sina Temesgen Tolera, Tesfaye Gobena, Nega Assefa, Abraham Geremew, Elka Toseva.

**Project administration:** Sina Temesgen Tolera, Tesfaye Gobena.

**Resources:** Sina Temesgen Tolera.

**Software:** Sina Temesgen Tolera, Tesfaye Gobena.

**Supervision:** Sina Temesgen Tolera, Tesfaye Gobena, Nega Assefa, Abraham Geremew.

**Validation:** Sina Temesgen Tolera, Tesfaye Gobena, Nega Assefa.

**Visualization:** Sina Temesgen Tolera, Tesfaye Gobena, Nega Assefa.

**Writing – original draft:** Sina Temesgen Tolera, Tesfaye Gobena, Nega Assefa, Abraham Geremew, Elka Toseva.

**Writing – review & editing:** Sina Temesgen Tolera, Tesfaye Gobena, Nega Assefa, Abraham Geremew, Elka Toseva.

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
