## [Decision Letter · Decision Letter 0]

30 Apr 2024

PONE-D-23-41265Burden and determinants of occupational injuries among sanitary workers in public hospitals, eastern EthiopiaPLOS ONE

Dear Dr. Tolera,

Thank you for submitting your manuscript to PLOS ONE. After careful consideration, we feel that it has merit but does not fully meet PLOS ONE’s publication criteria as it currently stands. Therefore, we invite you to submit a revised version of the manuscript that addresses the points raised during the review process. There are concerns in how you analyzed data, with logistic regression a poor choice for common outcome in a cross-sectional study. Fortunately, the remedy is simple as per my comments below.  Presentation of too many significant figures detracts from usefulness and readability: I make specific suggestions to address this in my comments below. Please also consider how much you gain by using p<0.05 for variable selection: it is not method of model-building that is favored for observational studies. Do your results change in any important way if you do not exclude any variables from regressions but simply present all effect estimates and then discuss their magnitude and precision of their estimates? 

We look forward to receiving your revised manuscript.

Kind regards,

Igor Burstyn

Academic Editor

PLOS ONE

3. We note that your Data Availability Statement is currently as follows: [All relevant data are within the manuscript]

4. Please amend the manuscript submission data (via Edit Submission) to include author Tesfaye Gobena, Nega Assefa, Abraham Geremew and Elka Toseva.

Additional Editor Comments:

Please revise your article using detailed feedback from reviewers as well as my comments that follow.

1.Please include copy of all questionnaires – in original language -- that you used as supplementary materials.

2.Translate key questions that you asked, such as about the outcomes and report in text; Saying that Boolean logic was used is both not adequate and not necessary if you give actual question and response options.

3.Report all percentages to with precision of 1/10, not 1/100, such that 97.29% should be replaced with 97.3%/ Make correction in text and tables. You simply do not have information with precision of 1/100 of 1% when you have <1000 subjects.

4.Report all COR and AOR (adjusted and unadjusted odds ratios) within 1 decimal place: e.g., 3.6, not 3.61. Make corrections to text and tables throughout.

5.For “mean ± SD for age, work 41 experience, educational status, and monthly income” please report precision within one decimal place or less.

6.You cannot conclude that “In order to reduce these risks, appropriate work-related policy relating 53 occupational health and safety be enacted. Further, enforce the implementation of occupational health 54 and safety practices within hospitals is necessary.” The reason for this is that you did not evaluate any interventions. Please remove all such claims from conclusions but feel free to speculate on plausible interventions in the discussion, esp. if there is evidence to support some specific interventions.

7.Among keywords, “determinants” is too vague. Determinants of exposure, outcome, something else? Please revise.

8.Table 3: round up all p-value to within 1 significant figure. Do not highlight values <0.05 with a *, because it is not appropriate to stress null hypothesis testing in observational studies, where non-random errors dominate.(Wasserstein and Lazar 2016)

9.The rate of self-reported occupational injuries was rather high, 44%, which makes logistic regression a poor tool for approximating relative risk. For OR to approximate RR, the rate of events should be in the range of 10% or less. A more appropriate method of data analysis is to estimate prevalence ratios (PR) using any number of available approaches. In STATA, modified Poisson regression (Zou 2004)is a sensible (better) method. I strongly recommend all analyses be re-done using PR. Please read this for description of implementation of the recommended analytical approach in STATA: https://stats.oarc.ucla.edu/stata/faq/how-can-i-estimate-relative-risk-using-glm-for-common-outcomes-in-cohort-studies/.

References

Wasserstein, R. L. and N. A. Lazar (2016). "The ASA's Statement on p-Values: Context, Process, and Purpose." The American Statistican 70: 129-133.

Zou, G. (2004). "A modified poisson regression approach to prospective studies with binary data." Am.J.Epidemiol. 159(7): 702-706.

Reviewers' comments:

Reviewer's Responses to Questions

**Comments to the Author**

1. Is the manuscript technically sound, and do the data support the conclusions?

Reviewer #1: Yes

Reviewer #2: Partly

Reviewer #3: Yes

2. Has the statistical analysis been performed appropriately and rigorously? 

Reviewer #1: Yes

Reviewer #2: Yes

Reviewer #3: Yes

3. Have the authors made all data underlying the findings in their manuscript fully available?

Reviewer #1: Yes

Reviewer #2: Yes

Reviewer #3: Yes

4. Is the manuscript presented in an intelligible fashion and written in standard English?

Reviewer #1: Yes

Reviewer #2: No

Reviewer #3: No

5. Review Comments to the Author

Reviewer #1: Reviewer comments

Manuscript Number = PONE-D-23-41265

Title: Burden and determinants of occupational injuries among sanitary workers in public hospitals, eastern Ethiopia

Dear authors, you need to address and clarify the following comments before publication

Abstract, line 33 delete from after sample size 809

Introduction

Line 60, what do you mean collect solid garbage? Is there liquid garbage?

Method

Line 117 what is mean Out of 5680 hospital staff of 568, about 809 of these were cleaners and waste collectors? What is your justification to take all sanitary workers as a sample? Why you no need sample among sanitary workers?

Line 135, If you took all sanitary workers, what is the need of sample size determination part?? Justify it?

Line 121. The map of your study area needs scale, and proper naming of figure is required

Line 151. 2.5 selection procedure. There is no proportional allocation, just you took the whole population? What is the need of stating sample selection?

Line 153, they are not selected?

Line 172 your study period was from May 30th, 2023 to August 30th, 2023, why the title of figure 2 is 2022?

Line 175 unfinished sentence

Line 181. 2.6.2 Independent Variables, what is the difference between Institutional variables and Working environment variables? May be enabling variables?

Line 202 contents of the questionnaire not question?

Line 208 the 7 days Boolean logic questionnaire not mentioned in your abstract of line 34

Line 210, what are Determinants questions: eighteen standard questions were prepared for associated factors mean? Are they different from other independent variables?

Line 228 Key informant interview = Is it data collection or analysis part? Justify why you need to summarize like quantitative data?

Line 247. What are the languages used for data collection?

Line 262, what is MRL?

Line 282 Table 1 Sociodemographic table: Is there separate division of labour for cleaners and waste collectors in the Ethiopian hospital human resource structure? Waste collectors from municipality or have their own?

Line 288, Burden of occupational Injuries, the calculation and narration is not clear?

Line 310, where do you get neutral attitude?

Table 3 and Table 4, If you ran logistic regression why you need to calculate X2 for comparison, please justify?

Table 4, Variables, Sex, Had MSD and conduct supervision are they significant?

Line 321. How do you measure Knowledge, attitude and practice as shown in your Table 4?

Line 330. 3.6 Assessment from IPC as KII, the analysis and result for Key Informant Interview is not clear please make it clear as qualitative data finding?

Line 465, strength, the study had no sample, instead all populations assessed

Line 480, Conclusion. You don’t have a variable indicating unsafe hygiene, how do you conclude in the first sentence?

Line 484. Have you addressed the rules , policies and regulations related to OHS?

Thank you

Reviewer #2: REVIEWER’S COMMENTS

Abstract

• Line 33: “Data were collected from 809 from hospital sanitary workers” should be read as “Data were collected from 809 hospital sanitary workers”

• Line 34: The occupational injury….., “The” should be deleted.

• Lines 36 – 38: “Binary and multiple logistic regressions were used to explore the relationship between dependent and independent variables. The crude odds ratio (COR) and adjusted odds ratio (AOR) with a 95% confidence interval (CI) were used to present the analysis.” Binary logistic regression reports crude ratio and multiple logistic regression also reports adjusted odds ratio so it is not necessary to include “The crude odds ratio ……..”. Authors can add “Statistical significance was determined at a 95% confidence level” after “…………. “between dependent and independent variables”.

• Line 39: Authors indicated in the methods section of the abstract that 809 respondents were used for the study. However, in the results section of the abstract, they indicated 729. The authors should clarify the inconsistency.

• Lines 41 – 42: Authors should indicate the unit of measurement of age and work experience.

• The results section is not very clear. Authors should consider re-writing the results to make them clearer.

Introduction

• Line 65: “They often cited” should be written as “They are often cited”.

• Lines 69 – 70: This needs to be revised to make it clearer.

• Line 73: “but also they are facing with a variety of…” should be written as “but also they face several occupational ….”.

• Line 97: Reference (35) should be in square brackets, [35].

Methods

• Line 109 – 111: “In terms of study areas, public hospitals within a 200km radius of Harar were chosen from three regions and one administrative city”. This statement is not clear. The authors should justify why they selected the study areas.

• Lines 114 – 117: “From …..” the content should be checked and it is a bit unclear.

• Line 115: The word “daily” should be deleted.

• Lines 116 – 117: The (mean+SD) values in the bracket are 4 but the parameters are three (bed occupancy, outpatient flow and inpatient). This should be rectified.

• Line 117: “Out of 5680 hospital staff of 5680” This statement should be checked as 5680 is repeated.

• The map of the study area is blurred. Authors should replace it with a clear map.

• Line 122: Section 2.2 “Source and study population” should be revised to read “Study population”.

• Line 134: The sentence is incomplete.

• Line 135: Section 2.4 - Sample determination. The entire section is not clear. Authors should be specific on how they estimated the sample size for the study.

• Line 151: Section 2.5 Selection Procedures should be written as “Sample selection procedures”. Authors should explain how they arrived at the sample size for each of the four study States.

• Line 165: About Figure 2. According to the information in section 2.5, Dire Dawa City has a total sample size of 82. However, it says otherwise in the Figure. The same applies to the Oromia Regional state.

• Line 175: “………workers’ representatives (KII-2) were purposely selected from each eight”. The authors should check this statement again as there may be an error with the use of “eight”.

• Lines 192 – 194: Authors should check the time shifts as they overlap and create confusion for readers.

• Lines 228 – 246: The information presented here does not form part of quantitative data. Key informant interviews are ways of collecting qualitative data. Authors should kindly revise.

• Line 258: “independent variable” instead of "variable independent”

• Lines 267 – 274: Section 2.10 Ethical statement. Authors should take the entire section out.

Results

• Lines 288 – 289: “Of these, 43.59% (95%CI: 39.87, 47.42) and 50.00% (95%CI: 35.52, 64.47) were cleaners and solid waste collectors, respectively”. The percentages of cleaners and solid waste collectors should sum up to 100%. This needs to be cross-checked.

• Table 3: Authors should provide a note under the table to explain the meaning of values with single asterisks.

• Line 330: Section 3.6 – Assessment from IPC as KII. This section per the methods is supposed to be qualitative. However, nothing shows under the results section. Authors should present their qualitative findings.

Discussion

• The discussion could be improved by revising the qualitative aspect to reflect the suggested revision in the results section.

• Authors should only discuss the findings based on the objectives of the study and delete any other thing that is not related to the objectives of the study.

• The authors could improve the discussion by including the implications of the findings.

Conclusion

• This will have to be reconsidered after effecting all the suggested corrections.

General comment

• There are many errors in the manuscript that need to be corrected. I suggest authors proofread the entire manuscript. Some parts have been highlighted yellow and green and I do not know the rationale for doing that. Authors should ensure they submit a clean document.

• A few mentions of the errors can be found in lines 53, 102, 103 115, 138 – 140, 192, 248 – 249, 256, 472, 473 – 474, and 486 – 487.

Reviewer #3: Abstract:

In the methods, include the sampling technique and data collection methods. Include the statistical analysis in a concise manner.

The results should include the factors found to increase the risk of occupational injuries. IPC should be provided in its full form.

Conclusions should be concise and based on important findings. Avoid making comments on policy implications that are not based on the findings.

Methods:

The purpose of mentioning mean, range, and standard deviation (mean ± SD) by bed occupancy (lines 114 to 117) is unclear.

The sentence “Out of 5680… and waste collectors” (lines 117-118) is not clear.

In the inclusion criteria, it is mentioned that newly recruited SW working for more than a month were included in this study. This would bias the results because the occupational injuries that occurred within the last 12 months influence the findings of prevalence by the respondents who joined as SW for more than a month.

Sample size determination: p-value for sample size calculation, p-value for age and PPE has been considered which is not appropriate, these should be avoided.

There is no sampling technique to include respondents

There should be criteria for selecting KIs (lines 175 to 177)

In independent variables (2.6.2), variable related to stress symptoms should be mentioned.

Results:

The results indicate that 729 out of the 809 calculated samples were respondents. It's important to explain why others did not respond.

In Table 2, there is a mistake in the total number of categories in the columns.

Limitations:

There may be a recall bias in reporting occupational injuries in the last 12 months; how this has been addressed should be mentioned.

Conclusions:

The comments (lines 484–488) regarding policy implications are not supported by the findings; however, these can be discussed in discussion.

6. PLOS authors have the option to publish the peer review history of their article (what does this mean?). If published, this will include your full peer review and any attached files.

Reviewer #1: **Yes: **It is my review

Reviewer #2: No

Reviewer #3: No

---

## [Author Response · Author response to Decision Letter 0]

7 Jun 2024

June, 2024

Subject: Sending revised manuscript, “PONE-D-23-41265” 

Greetings to 

PLOS ONE Editor

PLOS ONE Editorial team 

Reviewers

Kindly, as corresponding Author, on behalf of the rests, I would start by extending a warm greeting to the PLOS ONE’s chief editor, the editorial team, and the reviewers. Please allow me to express my gratitude to the PLOS ONE editors and reviewers for your sincere advice, criticisms, suggestions and recommendations. These contributions and efforts are very significant to the quality of our main manuscript in the current version. 

Having this, we have used the following format of table to address all issues forward from reviewers (Reviewer 1 and Reviewer 2) ,which embedded in the attached table. Within a manuscript, we used or highlighted with two colors: “Yellow” used for modification and “Cyan Green” used for newly added paragraphs addressed in “Revised Manuscript with Track Change”.

Some comments on Knowledge , Attitude, Personal protective equipment and Infection prevention questions were raised. For the mentioned items, the questions were prepared as associated /determinants of occupational injuries, which are submitted as supplementary material. The response for each categorical variables designed to use for other, despite of fact that we used them as summary by giving cut point for each variable., which explained in the method section. All suggestions and comments attached as "Response to Reviewers" through destination of submission platform.

Sincerely

Corresponding Author

---

## [Editor Report · Decision Letter 1]

12 Jun 2024

PONE-D-23-41265R1Burden and determinants of occupational injuries among sanitary workers in public hospitals, eastern EthiopiaPLOS ONE

Dear Dr. Tolera,

Thank you for submitting your manuscript to PLOS ONE. After careful consideration, we feel that it has merit but does not fully meet PLOS ONE’s publication criteria as it currently stands. Therefore, we invite you to submit a revised version of the manuscript that addresses the points raised during the review process.

We look forward to receiving your revised manuscript.

Kind regards,

Igor Burstyn

Academic Editor

PLOS ONE

Additional Editor Comments:

Authors do not seem to have submitted point-by-point response to reviewers. Please create a separate document which you respond in detail to every comment of 3 reviewers and academic editor. You can use the same format as your "To Editor1.docx". In each of these point-by-point responses state if you agree or disagree with the comment. If you disagree, explain why. If you agree, state how the manuscript was altered. It is not enough to simply color-code the changes, because this skips the essential contextualization of why you are making changes.  TO DO THIS GO BACK TO COMMENTS ON THE ORIGINAL SUBMISSION AS THE ARE CURRENTLY ARE NOT DEEMED AS ADDRESSED DUE TO LACK OF POINT-BY-POINT RESPONSE TO THEM FROM YOU.

Please also address the following critique:

1. As per pervious comments from me, delete all "Asterisk shows statistically significant at *p-value<0.001; 350 **p<0.01; ***p<0.05, ****p <0.20; *****p>0.20" and show exact p-values instead, rounded up to 1 significant figure in Tables (Table 3, Table 4. Do not cite any p-values or "statistical significance" in text.

2. In Table 5, p-values cannot be >1, because p is a probability lies between 0 and 1, included, by definition.

---

## [Author Response · Author response to Decision Letter 1]

16 Jun 2024

16 June, 2024

Subject: Sending revised manuscript, “PONE-D-23-41265” [Separate file also attached for you) 

Greetings to 

==>Reviewer #1

==>Reviewer #2

==>Reviewer #3

All Kindly, as corresponding Author, on behalf of the rests, I would start by extending a warm greeting to the reviewer 1, Reviewer 2 and Reviewer 3. Please allow me to express my gratitude to you for your sincere advice, criticisms, suggestions and recommendations. These contributions and efforts are very significant to the quality of our main manuscript in the current version. 

Dear Having this, we have used the following format of table to address all issues forward from you ,which embedded in the attached table. Within a manuscript, we used or highlighted with two colors: “Yellow "used for modification in line to your comments and suggestions and also “Cyan Green” used for newly added paragraphs to rich the manuscript and keep the quality , both these addressed in “Revised Manuscript with Track Change”. We conceptualized the feedback within the table why we changed or modified based on your comments. Separate file also attached for your feedback. Despite let us reflect the your comments, suggestion and recommendation raised from Reviewer 1 to Reviewer 3 step-by-step.

Some comments on Knowledge , Attitude, Personal protective equipment and Infection prevention questions were raised from you and other . For the mentioned items, the questions were prepared as associated /determinants of occupational injuries, which are submitted as supplementary material. The response for each categorical variables designed to use for other, despite of fact that we used them as summary by giving cut point for each variable., which explained in the method section . Dear due to entire correction, some of line numbers are may move up or down. Thus, we expect to politely understanding this problems while you scan the comments or suggestion thought the manuscript. 

Response for Reviewer #1

Introduction

Line 60, what do you mean collect solid garbage? Is there liquid garbage? We corrected as “solid waste”

Method

Line 117 what is mean Out of 5680 hospital staff of 568, about 809 of these were cleaners and waste collectors? What is your justification to take all sanitary workers as a sample? Why you no need sample among sanitary workers? We used as background of the study. 5680 staffs are those working in selected public hospitals including sanitary workers. But, our plan was only for those staffs their works are associated with cleaning and waste disposing . These are accounted 809 . Despite the statement is improved 

Line 135, If you took all sanitary workers, what is the need of sample size determination part?? Justify it? This section is revised based mother document (Dissertation). Thus, the design effect =2 was added in this version 

Line 121. The map of your study area needs scale, and proper naming of figure is required The map is changed considering city where the study areas (hospitals) found 

Line 151. 2.5 selection procedure. There is no proportional allocation, just you took the whole population? What is the need of stating sample selection? This section was fully revised including entire figure 2 based on suggestion (line 144-149)

Line 153, they are not selected? 

Line 172 your study period was from May 30th, 2023 to August 30th, 2023, why the title of figure 2 is 2022? Thank you, its corrected “2023”

Line 175 unfinished sentence Modified (Line 224-226)

Line 181. 2.6.2 Independent Variables, what is the difference between Institutional variables and Working environment variables? May be enabling variables? We revised this section and added both together (Line 207-214)

Line 202 contents of the questionnaire not question? Thank you, it modified as suggested (line 198)

Line 208 the 7 days Boolean logic questionnaire not mentioned in your abstract of line 34 It is added per as suggestion (line 39)

Line 210, what are Determinants questions: eighteen standard questions were prepared for associated factors mean? Are they different from other independent variables?

In our work, we used Determinants or independent variables or associated factors exchangeable.. Now, it changed into “Associated factor”

Line 228 Key informant interview = Is it data collection or analysis part? Justify why you need to summarize like quantitative data? Thus, we used some of them were for quantitative and s some of them qualitative. As you see some data converted into quantitative in order to compiled the data for easily understanding. 

Line 247. What are the languages used for data collection? Three languages: Somali, Oromo and Amharic, which are stated in data quality local language (Line 263-264)

Line 262, what is MLR?Thank you:MLR meant that multiple logistic regression, but In this version it was changed into modified Poison regression. It has been changed based genuine comments from PLOS editor. Really, we were happy on it and because it is milestone for our work because the major weakness of logistic regression is that it tends to over-estimate the risk if the outcome of interest is not rare. However, modified Poisson regression model is preferred in cross-sectional studies when the outcome of interest is not rare because it approximates the risk ratios or relative ratios better than binary logistic regression model (Line 292-296) 

Line 282 Table 1 Sociodemographic table: Is there separate division of labour for cleaners and waste collectors in the Ethiopian hospital human resource structure? Waste collectors from municipality or have their own? Of course, yes. Waste collectors are recruited in some of hospitals due to large of solid waste produced in hospitals. 

Line 288, Burden of occupational Injuries, the calculation and narration is not clear?

It was done as proportionally using their own denominator, that means 296/679 equals to 43.59% and 25/50 equals to 50.00% occupational injuries were cleaners and solid waste collectors, respectively.

By avoiding the above figures, it has been corrected as 

1)92.21% was for only cleaners in the hospitals

2)7.79 % was for only waste collectors in the hospitals 

3)321/721 (44.03% ) was both of them

Line 310, where do you get neutral attitude?

Attitude toward workplace risk was measured by likert scale (Assessment questions attached as supplementary material). The items then classified unfavored if socre <median(3.00) , neutral if score is 3.0 and favoured media is >3.00 (Line 210-213)

Table 3 and Table 4, If you ran logistic regression why you need to calculate X2 for comparison, please justify?

Table 3: Research team also agreed to delete Table 3 due to less significant for the readers 

Table 4, Variables: Had MSD and conduct supervision are they significant? According TO current analysis, those had MSD is contribution for likelihood of occupational Injuries at un-dajusted prevalence ratio(Line 322) while, lack of supervision had more likely to increase the risk of occupational injuries at adjusted prevalence ratio(line 336)

Line 321. How do you measure Knowledge, attitude and practice as shown in your Table 4? Knowledge, attitude and practice were measured with standard tools. All cut point of other tools are illustrated (Line 207-234)

Line 330. 3.6 Assessment from IPC as KII, the analysis and result for Key Informant Interview is not clear please make it clear as qualitative data finding? Thank you. KII well explained in line 236-256. Regarding their feedback ,some of KII responses were changed into quantitative data in order to summarize rather explaining for each KII. Thus, average of their feedback was used and the key information was narrated for the readers (Line 357-370). 

Line 465, strength, the study had no sample, instead all populations assessed

Thank you very much it is corrected as suggestion (Line 547)

Line 480, Conclusion. You don’t have a variable indicating unsafe hygiene, how do you conclude in the first sentence? Thank you: It modified (Line 562-567)

Line 484. Have you addressed the rules, policies and regulations related to OHS? Thank you for your information. We removed from conclusion and added in the discussion with appropriate meaning (Line 391-393).

6 Do you want your identity to be public for this peer review?  Reviewer #1: Yes , it is my review 

Once again thank you the honesty and put your huge effort in our work 

Dear, Dr./Prof.

Please come forward if you are not satisfied with our input so far in your comments or suggestions ; we will take it from you and make the necessary corrections. 

Response for Reviewer #2

Line 33: “Data were collected from 809 from hospital sanitary workers” should be read as “Data were collected from 809 hospital sanitary workers”. Thank you very much, it is corrected per a suggestion (line 35-37) 

Line 34: The occupational injury….., “The” should be deleted. Thank you it is corrected (Line 39)

Lines 36 – 38: “Binary and multiple logistic regressions were used to explore the relationship between dependent and independent variables. The crude odds ratio (COR) and adjusted odds ratio (AOR) with a 95% confidence interval (CI) were used to present the analysis.” Binary logistic regression reports crude ratio and multiple logistic regression also reports adjusted odds ratio so it is not necessary to include “The crude odds ratio ……..”. Authors can add “Statistical significance was determined at a 95% confidence level” after “…………. “Between dependent and independent variables”. Thank you very much: This statement is modified per current analysis type.(Line 38-42) 

Line 39: Authors indicated in the methods section of the abstract that 809 respondents were used for the study. However, in the results section of the abstract, they indicated 729. The authors should clarify the inconsistency. Thank you: 809 are all study units or what we planned But 729 are the only responded the questionnaires. The reason for the non-response participants were explained (Line 386-391)

Lines 41 – 42: Authors should indicate the unit of measurement of age and work experience. Thank you. But the research team agreed to delete sociodemographic variable due less important for the readers in the abstract section 

The results section is not very clear. Authors should consider re-writing the results to make them clearer. Thank you it is rewrite it as you, other reviewers and research team agreed.

Introduction

 Line 65: “They often cited” should be written as “They are often cited”.

Thank you it is inserted (Line 66)

Lines 69 – 70: This needs to be revised to make it clearer. Modified (Line 68)

Line 73: “but also they are facing with a variety of…” should be written as “but also they face several occupational ….”. Modified (Line 74-84)

Line 97: Reference (35) should be in square brackets, [35]. Thank you it was mistakenly cited by manually and it is correct in this version by ENDNOTE 

Methods

 Line 109 – 111: “In terms of study areas, public hospitals within a 200km radius of Harar were chosen from three regions and one administrative city”. This statement is not clear. The authors should justify why they selected the study areas. These statements are corrected base on what we done (Line 113-117)

Lines 114 – 117: “From …..” the content should be checked and it is a bit unclear. Modified and detailed (Line 117-120)

Line 115: The word “daily” should be deleted. Thank you it is deleted 

Lines 116 – 117: The (mean+SD) values in the bracket are 4 but the parameters are three (bed occupancy, outpatient flow and inpatient). This should be rectified. Thank you very much they separately stated in (Line 121-122)

 Line 117: “Out of 5680 hospital staff of 5680” This statement should be checked as 5680 is repeated.

Thank you , it is revised [Line 121-122]

The map of the study area is blurred. Authors should replace it with a clear map. It is modified with new one (Line 124)

Line 122: Section 2.2 “Source and study population” should be revised to read “Study population”. Thank you it is corrected (Line 125)

Line 134: The sentence is incomplete. Thank you both inclusion and Exclusion criteria were revised and rewrote (129-135)

Line 135: Section 2.4 - Sample determination. The entire section is not clear. Authors should be specific on how they estimated the sample size for the study.

This sectional is modified well after team discussion (Line 136-144)

Line 151: Section 2.5 Selection Procedures should be written as “Sample selection procedures”. Authors should explain how they arrived at the sample size for each of the four study States.

This sectional is also modified well after team discussion (Line 144-149)

Line 165: About Figure 2. According to the information in section 2.5, Dire Dawa City has a total sample size of 82. However, it says otherwise in the Figure. The same applies to the Oromia Regional state.

As justified under method section, two hospitals were randomly selected from four of the regions by providing equal change for each. Based on this Figure 2 is modified as the suggestions (Line 144-149)

Line 175: “………workers’ representatives (KII-2) were purposely selected from each eight”. The authors should check this statement again as there may be an error with the use of “eight”. Thank you. Corrected as from 8 hospitals, 8 IPC experts (KII-1) and 8 Sanitary representatives/Coordinators (KII-2) were used as KII(Line 136-138).. KII well explained in line 236-256. Regarding their feedback ,some of KII responses were changed into quantitative data in order to summarize rather explaining for each KII. Thus, average of their feedback was used and the key information was narrated for the readers (Line 357-370)

Lines 192 – 194: Authors should check the time shifts as they overlap and create confusion for readers. It means job rotation for consecutive one month, now it is corrected

Lines 228 – 246: The information presented here does not form part of quantitative data. Key informant interviews are ways of collecting qualitative data. Authors should kindly revise.

A lot of questions supported by closed and open for their opinion. Thus, both are included in narrative as well as quantitative based on adapted questionnaires 

Line 258: “independent variable” instead of "variable independent”

Thank you it is corrected (Line 181)

Lines 267 – 274: Section 2.10 Ethical statement. Authors should take the entire section out.

Editor suggested “Ethical statement” in method section (Line 292)

Results

 Lines 288 – 289: “Of these, 43.59% (95%CI: 39.87, 47.42) and 50.00% (95%CI: 35.52, 64.47) were cleaners and solid waste collectors, respectively”. The percentages of cleaners and solid waste collectors should sum up to 100%. This needs to be cross-checked. It was proportionally reported from the following table 

But, now it is corrected within the manuscript based on the following figure 

Table 3: Authors should provide a note under the table to explain the meaning of values with single asterisks. We did it even in the current version under each table. Despite The previous Table 3 is removed due less significant in our work. We replaced it with Modified Poisson Regression Bivariate analysis of out put 

 Line 330: Section 3.6 – Assessment from IPC as KII. This section per the methods is supposed to be qualitative. However, nothing shows under the results section. Authors should present their qualitative findings. Thank you very much.KII recruited for both quantitative and some qualitative information. Therefore, almost the important issues regarding our findings are addressed in this version (Line 377-394)

Discussion

The discussion could be improved by revising the qualitative aspect to reflect the suggested revision in the results section.

Thank you. The discussion and result section have been improved as editor and all of you suggested . 

 Authors should only discuss the findings based on the objectives of the study and delete any other thing that is not related to the objectives of the study. Thank you for the feedback. In this version, we in lighted to only the scope of study including qualitative evidence in order to rich the quantitative evidence 

 The authors could improve the discussion by including the implications of the findings.

---

## [Editor Report · Decision Letter 2]

23 Jun 2024

PONE-D-23-41265R2Burden and determinants of occupational injuries among sanitary workers in public hospitals, eastern EthiopiaPLOS ONE

Dear Dr. Tolera,

Thank you for submitting your manuscript to PLOS ONE. After careful consideration, we feel that it has merit but does not fully meet PLOS ONE’s publication criteria as it currently stands. Therefore, we invite you to submit a revised version of the manuscript that addresses the points raised during the review process.

We look forward to receiving your revised manuscript.

Kind regards,

Igor Burstyn

Academic Editor

PLOS ONE

Additional Editor Comments:

Please respond point by point to my comments that were at the end of last decision letter. You already responded to 3 reviewers (thanks!) but you also MUST respond to my comments, editor's. Once all this is done, your revision will be reviewed otherwise it may be rejected for failure to follow instructions.

---

## [Author Response · Author response to Decision Letter 2]

1 Jul 2024

July, 2024

Subject: Sending revised manuscript, “PONE-D-23-41265 R2” 

Greetings to 

==>PLOS ONE Academic Editor

==>All reviewers

Kindly, as corresponding author, on behalf of the rests, I would start by extending a warm greeting to the PLOS ONE’s Academic editor, Reviewers as well as editorial team. 

**

Please allow me to express my gratitude to the PLOS ONE academic editors and all reviewers for your sincere advice, criticisms, suggestions and recommendations. These contributions and efforts are very significant to the quality of our main manuscript in the current version. 

**

Dears! almost all comments were acceptable , thus we can say that almost are addressed. In some comment few justifications have been reflected to PLOS ONE Academic and for the reviewers regarding to your/their comments and correction. 

**

Thus, we prepared response for reviewers labeled as “Response to Reviewer” for Academic editor and the reviewers : Accordingly, we used

1) "Response to Academic Editor" for Editor 

2) “Response to Reviewer 1” for Reviewer 1

3) “Response to Reviewer 2” for Reviewer 2

4) “Response to Reviewer 3” for Reviewer 3

**

Thus, four (1-4) separate files were attached in destination of Response to Reviewer on submission system.

Sincerely 

Corresponding Author

---

## [Decision Letter · Decision Letter 3]

20 Aug 2024

PONE-D-23-41265R3Occupational Injuries and Associated Factors among Sanitary workers In Public Hospitals , Eastern EthiopiaPLOS ONE

Dear Dr. Tolera,

Thank you for submitting your manuscript to PLOS ONE. After careful consideration, we feel that it has merit but does not fully meet PLOS ONE’s publication criteria as it currently stands. Therefore, we invite you to submit a revised version of the manuscript that addresses the points raised during the review process.

We look forward to receiving your revised manuscript.

Kind regards,

Igor Burstyn

Academic Editor

PLOS ONE

Journal Requirements:

**Additional Editor Comments:**

Thank you for extensive revisions.

Reviewers have some specific corrections and questions that they wish you to consider.

Furthermore, while you say that you included all the questionnaires in the original language, these do not appear to be in the file that was submitted for our review. Can you please ensure that it is included in the next version and reference such supplemental materials in text, where methods are described.

Reviewers' comments:

Reviewer's Responses to Questions

**Comments to the Author**

1. If the authors have adequately addressed your comments raised in a previous round of review and you feel that this manuscript is now acceptable for publication, you may indicate that here to bypass the “Comments to the Author” section, enter your conflict of interest statement in the “Confidential to Editor” section, and submit your "Accept" recommendation.

Reviewer #1: (No Response)

Reviewer #3: All comments have been addressed

2. Is the manuscript technically sound, and do the data support the conclusions?

Reviewer #1: Yes

Reviewer #3: Yes

3. Has the statistical analysis been performed appropriately and rigorously? 

Reviewer #1: Yes

Reviewer #3: Yes

4. Have the authors made all data underlying the findings in their manuscript fully available?

Reviewer #1: Yes

Reviewer #3: Yes

5. Is the manuscript presented in an intelligible fashion and written in standard English?

Reviewer #1: Yes

Reviewer #3: Yes

6. Review Comments to the Author

Reviewer #1: Dear authors,

It is a great improvment, and I fill some editorial and clarification commonts to be addressed for the better ment of the paper quality

Reviewer comments # 2

Title of manuscript: Occupational Injuries and Associated Factors among Sanitary Workers in public hospitals, eastern Ethiopia

Manuscript Number: PONE-D-23-41265R3

Abstract

Line 41: selected at randomly. Should be selected randomly or selected at random.8 (Eight data collectors)

And 8 should be written in letters as a starting of new sentence on the same line

Line 42. 809 when you start a sentence, please use the letter no number

Line 44. What is descriptive statistical?

Line 46. Bivariate and multivariate regression is used for more than one outcome variable.

What is the difference between bi-variable VS Bi-variate?

Line 62: The study also found that SWs those other acquired occupational diseases within hospitals, What is this?

Method and materials

Line 117: At stage 1st ??? at 1st stage or at stage one?

Line 118 Vs line41?? Then eight public hospitals were selected at random from a total of fourteen public hospitals with an equal probability allocated to each region. Vs Out of 16 hospitals, 8 of them were selected at randomly. What is the number of hospitals in the Eastern Ethiopia?

Line 129 figure one. The geographical coordinates should be indicated for the study area.

Line 148. 5% contingency is not the write term for study participants. Instead use nonresponse rate

Line 154-155 please make it clear the abbreviation part for hospitals?

Line 240. What is some key Quantitative and quantitative information from KII? I think KII is used to gather qualitative data to enrich the quantitative finding, please elaborate more?

Line 285: what is Accordingly, the value o was 0.93,?

Line 309: Please be consistent with data presentation in brackets (44.0% (95%CI: 40.4, 47.7). Of these, 92.2% (95%CI: 88.7,94.90%) and 7.8% (95%CI: 5.1, 11.3%). Either use % for all or leave it for all.

Table 3. Bi-variate ? VS Multi-variable in table 4, how do you relate the two tables? If your outcome variable is one, how you use multivariate analysis?

Thank you

Reviewer #3: Thank you, authors; you have addressed all my queries and comments satisfactorily.

However, Table 2 is too long; for better understanding, separate it into 'injury type', 'body part injured', and 'reasons for injuries'.

There is still scope for editing to further improve the English language.

7. PLOS authors have the option to publish the peer review history of their article (what does this mean?). If published, this will include your full peer review and any attached files.

Reviewer #1: No

Reviewer #3: No

---

## [Author Response · Author response to Decision Letter 3]

6 Sep 2024

September, 2024

Subject: Sending revised manuscript, “PONE-D-23-41265” 

I have warming Greetings to 

 � Reviewer #1

 � Reviewer #2

 � Reviewer #3

Kindly, as corresponding Author, on behalf of the rests, I would start by extending a warm greeting to the reviewer 1, 2 and 3 .Please allow me to express my gratitude to you for your sincere advice, criticisms, suggestions and recommendations. These contributions and efforts are very significant to the quality of our main manuscript in the current version.

** 

Dear Having this, we have used the following format of table to address all issues forward from you ,which embedded in the submitted on submission system under destination of “Respond to Reviewers “ separately. We also submitted two manuscripts: 1) Revised Manuscript with highlighted and Revised Manuscript(Clean document) . Revised Manuscript with highlighted with two colors: “Yellow "used for modification in line to your and other reviewers’ comments and suggestions and also “Cyan Green” used for newly added paragraphs to rich the manuscript and keep the quality in “Revised Clean Manuscript and Revised Manuscript with Track Change”. We also reflected your feedback within the table as well as comments that have been addressed. 

**

Kindly, Dear due to entire correction, some of line numbers are may move up or down. Thus, we expect to politely understanding this problems while you scan the comments or suggestion thought the manuscript. 

Sincerely

Corresponding Author

---

## [Editor Report · Decision Letter 4]

11 Sep 2024

Occupational Injuries and Associated Factors among Sanitary workers In Public Hospitals , Eastern Ethiopia: A Modified Poisson regression Model Analysis

PONE-D-23-41265R4

Dear Dr. Tolera,

We’re pleased to inform you that your manuscript has been judged scientifically suitable for publication and will be formally accepted for publication once it meets all outstanding technical requirements.

Kind regards,

Igor Burstyn

Academic Editor

PLOS ONE

Additional Editor Comments (optional):

Thank you for undertaking revisions and responding to comments. While modified Poisson regression is not particularly novel in health research, if you feel that you need to highlight it in the title, I trust that this would be helpful to researchers in your domain.

Reviewers' comments:

none.

---

## [Editor Report · Acceptance letter]

18 Sep 2024

PONE-D-23-41265R4 

PLOS ONE

Dear Dr. Tolera, 

I'm pleased to inform you that your manuscript has been deemed suitable for publication in PLOS ONE. Congratulations! Your manuscript is now being handed over to our production team.

Kind regards, 

on behalf of

Dr. Igor Burstyn 

Academic Editor

PLOS ONE